# Intradecadal variations in length of day and their correspondence with geomagnetic jerks

Pengshuo Duan [ID] [1✉] & Chengli Huang [ID] [1,2✉]

Earth's core oscillations and magnetic field inside the liquid outer core cannot be observed directly from the surface, we can infer these information from the intradecadal variations in Earth's rotation rate defined by length of day. However, the fine time-varying characteristics as well as relevant mechanisms of the intradecadal variations are still unclear. Here we report that the intradecadal variations present a significant 8.6-year harmonic component with an unexpected increasing phenomenon, besides a 6-year decreasing oscillation. More importantly, we find that there is a very good correspondence between the extremes of the 8.6-year oscillation with geomagnetic jerks. The fast equatorial waves with subdecadal periods propagating at Earth's core surface may explain the origin of this 8.6-year oscillation.

[1] CAS Key Laboratory of Planetary Sciences, Shanghai Astronomical Observatory, Chinese Academy of Sciences, Shanghai 200030, China. [2] University of Chinese Academy of Sciences, Beijing 100049, China. ✉email: duanps@shao.ac.cn; clhuang@shao.ac.cn

The intradecadal variation (i.e., 5–10-year scales) in length of day (LOD) is an interesting topic in fundamental astronomy and geophysics as it may closely correlate with the fast dynamics of the Earth's core[1,2] and the geomagnetic field changes[3–5]. The existence of a significant 6-year periodic oscillation with a mean amplitude of ~0.12 ms existing in the observed LOD data has been confirmed by many works[6–9], while the recent studies seem to have equated the intradecadal variations in LOD with the 6-year oscillation. However, the time-varying characteristics of the intradecadal variations in LOD[10,11] and the corresponding frequency-domain results from the Fourier spectral analysis[8] indicated that the intradecadal variations may contain more harmonic components than currently widely accepted thoughts (i.e., a 6-year signal alone), since the frequency domain result of the LOD variations does not present a single sharp 6-year peak on the 5–10-year scales.

In order to quantitatively detect the time-varying characteristics of the intradecadal variations in LOD, here we adopt a wavelet analysis method named normal Morlet wavelet transformation (NMWT)[12], which owns a high-frequency resolution to analyze the intradecadal variations (see "Methods" section and Supplementary Note 1), and the NMWT method is proved to be an effective approach to accurately recognize and quantitatively extract the target harmonic signals with close frequencies from the original data. It should be noted that when the wavelet method is used to accurately detect the real LOD data, a strategy of avoiding its edge effects (EE) needs to be designed. Given that the previous works[8,13] did not clearly illustrate the method of avoiding the EE in detail, here, in order to obtain accurate and robust LOD results and to make our work be repeatable easily, we adopt a simple strategy called as boundary extreme point mirror-image-symmetric extension (BEPME) to avoid the relevant EE (see "Methods" section).

Here we present two obviously harmonic components (i.e., 6-year and ~8.6-year terms) existing in observed LOD data on the intradecadal scales, where the 8.6-year signal shows an increasing trend in time domain, which is first found to closely associate with geomagnetic jerks. This observed evidence indicates that the 8.6-year signal and geomagnetic jerks may result from a same physical source, i.e., the equatorial quasi-geostrophic (QG) Alfvén waves focusing at the Earth's core surface. This work provides a new possible entry to predict the rapid geomagnetic field changes ahead and to study the magnetohydrodynamics of the Earth deep interiors via Earth rotation variations.

## Results

**Intradecadal variations and a 8.6-year increasing signal.** Figure 1 shows the observed LOD variations during 1962–2019, from which the atmospheric angular momentum (AAM) effect has been removed, while we apply the 12-month and 6-month running average methods to eliminate the remaining seasonal signals (i.e., the annual and semi-annual terms) of the LOD variations. Using the Daubechies wavelet fitting method following the refs. [8,13], in which this wavelet filtering cannot produce the Gibbs effect[13], we obtain the background trend (i.e., the red curve in Fig. 1a, which mainly presents the LOD variations with period $T > 10$ years) and the residual series (the green curve in Fig. 1a mainly reflects the intradecadal variations with periods of 5–10 years, see Fig. 1b), where the coincidence between this residual series and the original LOD data in frequency domain (Fig. 1b) shows that the residual series can well characterize the intradecadal variations in LOD.

Additionally, the residual series shows the so-called modulation phenomenon[7,10,11,14] of the LOD variations on the intradecadal scales, while the frequency-domain result of the original LOD data (Fig. 1b) shows a wide energy-spectrum range on the 5–10-year scales instead of a 6-year sharp peak, meaning

that the characteristics of the intradecadal variations in LOD does not present a 6-year oscillation alone. Furthermore, the NMWT spectrum (Fig. 1c, d) shows that the intradecadal variations are more complex than the current findings, i.e., on the intradecadal scales, besides a 6-year oscillation, an obvious ~8.6-year periodic signal and some relatively weak periodic signals between these two periods (i.e., the 6-year and the 8.6-year) are also presented in the time–frequency spectrum. The origin issues of these relatively weak signals are beyond the scope of this work, though they may reflect the signatures of the fluid outer core (FOC) motions[11,14], here we do not exclude a possibility that they may be the consequence of stochastic excitation from two free normal modes (i.e., the 6-year and 8.6-year signals) (e.g., see the case 9 in Supplementary Fig. 19). More definitive conclusion still needs to be further explored later.

Whether this 8.6-year signal displayed in Fig. 1c, d is related to the removal of the background trend? Here, our simulation tests (Supplementary note 4) give a negative answer. Moreover, the Fourier spectral analysis of the original LOD data (the blue curve in Fig. 1b, where cpm refers to the abbreviation of cycles-per-month) also shows a wide energy-spectrum range within the 5–10-year band, which coincides well with that of the residual series, revealing the existence of the 8.6-year periodic component. In addition, a ~8.5-year peak in frequency domain of the LOD variations is also shown by a recent work[15], but the characteristic of this signal in time domain has never been shown. Then, why these two harmonic components (i.e., the 6-year and the 8.6-year terms) in LOD cannot be separated from each other by the traditional Morlet wavelet transformation (TMWT) spectrum[7,15] or other methods (e.g., the singular spectrum analysis (SSA), see "Methods" section)? This is mainly due to the issues of the frequency-resolution of these methods (see Supplementary Figs. 4–7). Therefore, we suggest that the modulation phenomenon mentioned above does not reflect the changes of the 6-year signal itself, but the result of the superposition of periodic harmonic components (e.g., the 6-year and 8.6-year periods), while the physical origins of these oscillation signals in LOD are interesting topic[1,6,11,14,16–20] and we will try to discuss them in this work.

Combining the NMWT method with the BEPME strategy, we can recognize the target 6-year and 8.6-year signals and extract them in time domain, respectively, and the results are shown in Fig. 2, which shows that their average amplitudes are respective ~0.124 and ~0.08 ms during 1962–2018. This work further confirms the phenomenon that the 6-year oscillation in LOD shows a secular decreasing trend[8,13] with an observed quality factor $Q\sim51$. Here, it should be noted that, based on the currently observed LOD data, we have not found the strong evidence to demonstrate that the observed 6-year oscillation in LOD (see Fig. 2a) has been undergoing the significant excitation during 1962–2019, since we have not found the relevant reliable stochastic excitation series (or events). Conversely, the current 6-year oscillation time-domain result can be well characterized by a free exponentially decaying function[18] (here, the exponential decaying factor $\beta$ is estimated to be ~$8.4 \times 10^{-4}$/month), hence a possible damping model of the 6-year oscillation was established by ref. [13]. Nevertheless, we do not exclude a possibility that this observed decaying phenomenon of the 6-year oscillation might be the consequence of a continuously stochastic excitation of a 6-year periodic normal mode (see "Methods" section and the Supplementary Fig. 16). Here, the attenuation and the excitation of the 6-year oscillation still need to be further studied in future using longer LOD data.

Differing from the 6-year decaying oscillation during 1962–2018, interestingly, we first find that the 8.6-year oscillation presents an unexpected long-term increasing trend (Fig. 2b), and

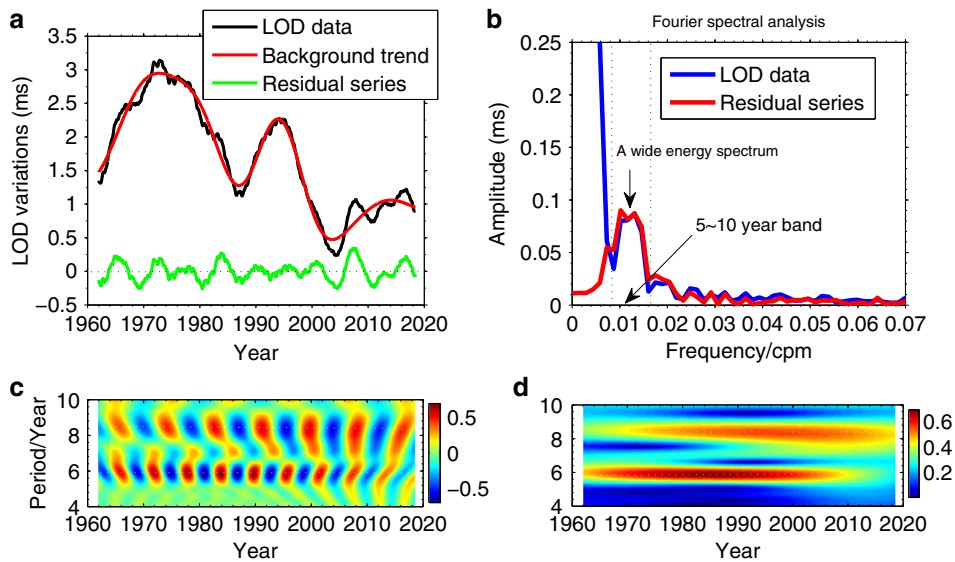

**Fig. 1 The variations of length of day in time frequency domain. a** Shows the LOD variations on the various scales, where the residual series (i.e., the green curve) mainly reflects the intradecadal variations, the frequency-domain result of which is shown in **b**; **b** presents the Fourier spectrum of the LOD data and the intradecadal variations, which shows a wide energy-spectrum range in 5–10-year band instead of a single 6 years sharp peak; **c**, **d** show the NMWT time–frequency spectrum, which further reveals the periodic components existing on the 5–10-year scales. Here, the window width factor $\sigma$ in NMWT method is set to be 3, which is large enough to clarify the target harmonic components, where the edge effect of the NMWT method and the strategy to eliminate it are illustrated in "Methods" section.

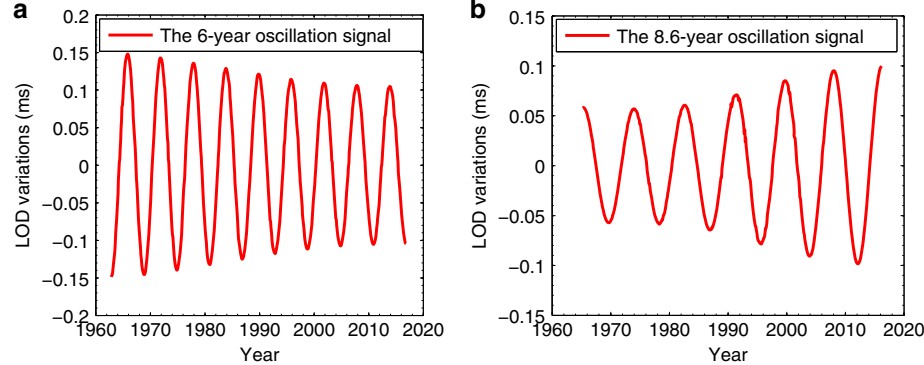

**Fig. 2 The 6-year and 8.6-year signals in time domain recovered by this work. a** Confirms that the 6-year oscillation is a decaying oscillation[13] with an observed quality factor ~51 (i.e., the currently observed decaying rate is about $8.4 \times 10^{-4}$/month); **b** shows that the 8.6-year signal presents an increasing phenomenon. Here, the phase information of these two signals is recovered accurately from the simulation analysis (Supplementary Figs. 2, 3, 9, 14 and 15). In this work, according to requirement of the BEPME strategy (see "Methods" section), the 6-year oscillation recovered is ended at 2016.4, while the 8.6-year signal is ended at 2016.0.

the 8.6-year amplitude-increasing phenomenon should be attributed to a possible continual excitation.

We further compare the above results with the original LOD variations, and the results are shown in Fig. 3. It shows that the composite signal (i.e., background trend +6 years +8.6 years) is in general consistent with the original LOD variations. This superposition signal (i.e., 6-year term +8.6-year term) can nicely characterize the general time-varying characteristics of the intradecadal variations in LOD, except some deviations from the original data (e.g., the periods of 1972–1974 and 2014–2016), which should be due to the disturbances from the other weaker signals existing on the residual series, this point can be shown in Fig. 1b–d. The general temporal-varying characteristics of the intradecadal variations in LOD can be explained by the super-position of a 6-year decaying signal and a 8.6-year increasing signal, which means that the temporal-varying characteristics of

the intradecadal LOD variations do not reflect the amplitude modulation of the 6-year signal itself, but the consequence of the superposition of (at least) the two signals (i.e., 6-year and 8.6-year components). In order to confirm the above results, we made some typical simulations (see Supplementary Note 4) to demonstrate the reliability of the whole LOD data processing involved in this work.

Although we can avoid the wavelet edge effect to a great extent through adopting the BEPME strategy (see "Methods" section), some deviations still exist near the boundaries of the data (see Supplementary Fig. 3), while removing the background trend may also cause some disturbances on the target intradecadal variations (see Supplementary Figs. 14 and 15), which, nevertheless, cannot influence the overall characteristics of the final results. Since the results recovered by the method proposed in this work are always slightly smaller than the actual value at the larger magnitude side

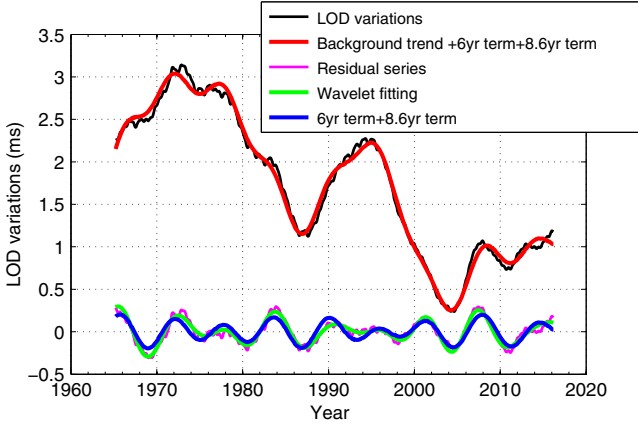

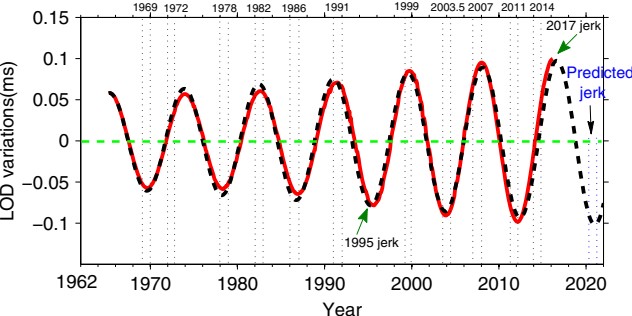

**Fig. 3 Comparisons of the recovered results and the original LOD variations.** In this figure, LOD data refers to the observed LOD data (from which the AAM effect has been removed, while a running average approach is managed to remove the remaining annual and semi-annual signals) as displayed in Fig. 1a; Background trend indicates the decadal variations presented in Fig. 1a. Residual series mainly reflects the intradecadal variations which is obtained from the Original data minus the Background trend.

**Fig. 4 Correspondence between the 8.6-year signal and geomagnetic jerks.** In this figure, the red curve expresses the recovered 8.6-year signal in LOD, while the black dashed curve shows the fitting result (i.e., an exponentially increasing model with the expression of $y(t) = A_0 \exp[\alpha(t - t_0)]\cos(2\pi f(t - t_0))$, where the initial amplitude $A_0 \approx 0.06$ ms; the currently observed exponential rate $\alpha \approx +0.00131$/month; $f \approx 0.00969$ cpm; the initial time $t_0$ is set to be at June 1982) of the red curve, which may be used to predict the time when the next new jerk (i.e., the predicted jerk in blue fonts) will probably happen.

of the original data (see Supplementary Figs. 3, 9, 14, and 15), the actual amplitude increasing of the 8.6-year signal should be slightly larger than our current result shown in Fig. 2, that is to say, the actual amplitude increment of the 8.6-year signal during the past several decades is perhaps somewhat underestimated by this work.

**Correspondence between the 8.6-year signal and geomagnetic jerks.** Why the amplitude of the 8.6-year oscillation in LOD shows a secular increasing trend during the past several decades? If this 8.6-year signal is attributed to the FOC torsional normal mode[14] (this is to be further discussed in the next section), then its amplitude increasing is possibly due to the excitation forcing within the FOC[14,21]. In addition, a geomagnetic jerk defined as the "V-shape" feature of the geomagnetic secular variations[2,22,23] essentially reflects a rapid change of the second-order time derivative of the geomagnetic field, which reflects changes of the shortest observable time scale of the Earth core field[24,25]. The idea that the geomagnetic jerks originating from the Earth interiors has been widely accepted[26–30], and the jerks were supposed to be closely related to the liquid flow motions at the surface of the FOC[28,29,31], which may associate with the angular momentum transfers between the core and the mantle[5,6,28,29]. Therefore, the jerk events may associate with the LOD variations on the intra-decadal scales[5,6,28]. Here, a scientific question arises, i.e., whether the amplitude increasing of the 8.6-year oscillation is related to the physical sources which can cause the jerks?

In this work, it is firstly found that there is a very good correspondence between the geomagnetic jerk timings with the extremes of the 8.6-year signal (Fig. 4): For example, all the following four well-known jerks[26,29] (i.e., 1969, 1978, 1991, 1999) well correspond to the extremes of the 8.6-year signal; moreover, we list the following seven jerks (1972, 1982, 1986, 2003.5, 2007, 2011, and 2014) from the previous works[6,25,32]. Given that the geomagnetic jerks are generally localized expressions at the Earth's surface, occasionally observed over large parts of the globe, which do not occur at the same time in all regions of the globe due to mantle conductivity[26,27], consequently, it is difficult to accurately define a single jerk time from the observations with the uncertainties (~±1 year) of the jerk occurrence and many

local secular accelerations overlap in time and space[23,33,34]. Therefore, the jerk epochs listed above may be not accurate enough, despite this, these epochs are regarded to be the best determinations[6].

Interestingly, Fig. 4 shows that almost all the above jerk timings coincide with the extremes of 8.6-year signal very well within ~1 year (or less). There are nine jerk epochs leading the extremes of the 8.6-year signal <1 year, except the 1972 jerk and 2014 jerk[32,35]. Here, the question that why these two jerks did not occur at the corresponding extremes of the 8.6-year signal are worthy to be discussed later. Besides, Fig. 4 also shows an absence of a 1995 jerk. Nevertheless, a potential jerk event was shown to occur around 1995 through analyzing the relation between free core nutation and jerks[36]. Meanwhile, another work[23] used the monthly mean geomagnetic data to discuss the geomagnetic jerk occurrence and find the jerk abound feature, where a jerk event happened during 1995–1998, though the jerk span almost fills the entire span at recent epochs[23]. The most recent SWARM satellite data[37] showed that a new jerk event might occur in 2017 (i.e., the 2017 jerk in Fig. 4) and this jerk event is also coincident with the extreme of the 8.6-year oscillation.

In summary, this phenomenon that the jerks closely correlates with the 8.6-year signal (see Fig. 4) provides an observed evidence to support the viewpoint that jerk occurrence may own a certain periodicity as the previous works[23,32,34,35] suggested, for instance, the jerk occurrence rate over the last several decades was suggested to occur at intervals ranging from 3 to 5 years[35], moreover, the jerk polarity changes were shown to own periodic characteristic[23], and mechanism response for the jerks may be related to a certain periodic oscillation[23,32,35].

**On the mechanisms and geophysical implications.** In this section, we will further discuss about the potential mechanisms of the intradecadal variations in LOD, especially clarifying the different physical origins of the 6-year and 8.6-year signals. From the intradecadal variations in LOD and their time-varying characteristics, one can infer the geomagnetic field strength inside the Earth's core[1,14], the information about the azimuthal torsional oscillation within the FOC[1,11,14], the core–mantle gravitational coupling strength[20,38], the electrical electricity at the lowermost mantle[6,13,18] and inside of the Earth's core[16], etc. The coincidence[11] of the predicted LOD variations from the ensemble

average torsional oscillation flow model and the observed LOD changes on 4–9.5 years reminds us that the fast torsional waves within the FOC may also have the corresponding 8.6-year periodic component found in this work, as the torsional waves can transfer angular momentum from the FOC to the mantle[14], and then cause corresponding LOD variations[11,14].

However, both the Fourier analysis of the LOD data (see Fig. 1b) and the simulation analysis (e.g., the Supplementary Figs. 5 and 6) indicate that the wide energy-spectrum of the 4–9.5 years variations in LOD obtained by the band-pass filtering[11,14,39] cannot be explained by a single 6-year oscillation, which just demonstrates the existence of other signal components besides the 6-year term, while the NMWT spectrum can further distinguish these components and reveal the presence of an 8.6-year signal clearly (i.e., Fig. 1c, d). In fact, the mechanisms of the 6-year and 8.6-year signals in LOD may be different, however, previous works have not well clarified the different geophysical origins of the intradecadal variations of the LOD.

As many published works indicated[16–18,20], one possible mechanism responsible for the 6-year oscillation in LOD is that the inner core (IC) swings with the 6-year eigenperiod under the action of the gravitational torque from the mantle, i.e., the mantle–IC gravitational coupling mode[17]. In this mode, although the angular momentum budget from the IC is small due to its much smaller inertia moment than that of the mantle, the observed amplitude of the 6-year oscillation is also not large (only ~0.12 ms). In other words, this small angular momentum budget from the IC is still large enough to explain the observed 6-year oscillation in LOD (see "Methods" section). Here, it should be noted that a partial FOC will be strongly coupled to the solid IC during this IC swing under the action of the electromagnetic coupling effects[40]. In this case, the FOC fast torsional oscillations with the 6-year recurrence period propagating from the IC to the equator at the CMB detected by ref. [1] is possibly due to this IC intrinsic swing under the action of the magnetohydrodynamics of the Earth deep interiors[40,41]. Nevertheless, there is yet no definitive scenario for their triggering, the Lorentz torques on the IC, or within the bulk of the FOC, appears to equally well generate waves traveling from the IC[14,21].

As to the 8.6-year oscillation in LOD. One possible mechanism for this oscillation is attributed to the fluid core torsional oscillation normal mode[11,14]. In this case, it will be not appropriate to use the observed 6-year period to estimate the cylindrical radial component of the magnetic field ($\tilde{B}_s(s)$) inside the FOC from the eigenperiod formula[1,14,42] of the FOC torsional oscillation normal mode. Instead the 8.6-year period should be used to infer $\tilde{B}_s(s)$ with the following formula[1,14,42]:

$$\tilde{B}_s(s) \approx \frac{r_f}{\tau}\sqrt{\rho_0\mu_0} \qquad (1)$$

where $\tilde{B}_s(s) = \sqrt{\frac{1}{4\pi h}\int_{-h}^{+h}\int_0^{2\pi}B_s^2(s,\phi,z)\,\mathrm{d}\phi\,\mathrm{d}z}$; here $h(s) = \sqrt{r_f^2 - s^2}$ is the half-height of a fluid cylinder, and the cylindrical $(s, \phi, z)$ coordinates is adopted, i.e., $s$ is the radius of the cylinder, $\phi$ expresses the longitude, $z$ direction is aligned with the Earth rotation vector $\vec{\Omega}$, $r_f (=3.48\times 10^6$ m) is the radius of the CMB, $\tau = 8.6$ years (the eigenperiod of the normal mode), $\rho_0 (=1.1\times 10^4$ kg m$^{-3}$) is the average density of the FOC, $\mu_0$ is the vacuum permeability with the value of $4\pi\times 10^{-7}$ H/m. According to the formula (1), we can estimate $\tilde{B}_s(s) \sim 1.5$ mT, which is consistent with the strong magnetic field strength within the FOC (1–4 mT) inferred from tidal dissipation[43].

Another alternative mechanism responsible for the 8.6-year signal in LOD is possibly due to the fast equatorial waves with the subdecadal periods propagating at the top of the FOC[24,25], which have been inferred by the current geomagnetic satellite (i.e., CHAMP and DMSP—Dense Meterological Satellite Program) data and ground observatory data[24], and these waves were suggested to be the normal mode signals (with the eigenperiod $T \sim 8.5$ years) of the secular acceleration of the fluid core motions[24], which seems to have characteristics of the magnetic Rossby waves in a stratified layer, though a most recent work[44] did not favor the presence of a stratified layer at the top of the outer core.

## Discussion

Considering the assumption[14,39,45] of significantly stochastic excitation forcing distributing within the bulk of the FOC, the normal mode signals existing in the LOD intradecadal variations (e.g., the MICG mode and the torsional oscillation normal mode) may be masked by the noises produced by the AR-1 stochastic process[14,39]. However, the question why the purely stochastic forcing distributes within the bulk of the FOC is retained and seems not to be easily answered. In addition, although it is difficult to time geomagnetic jerks accurately and to assess correlations between geomagnetic jerks and other phenomena, this work is just an effort in this respect. Through extracting a new harmonic component (i.e., 8.6-year signal) existing in LOD variations and discussion of its physical origin and its relations to geomagnetic jerks, we hope that this work can make an advance in finally solving this problem on the relationship between geomagnetic jerks and Earth rotation variations.

Combining the result of this work (Fig. 4) with a recent numerical simulation analysis[28], the 8.6-year signal in LOD and geomagnetic jerks may result from a same physical source, i.e., the so-called QG Alfvén waves focusing at Earth's core surface. Furthermore, the geomagnetic jerks can be induced by the arrivals of localized Alfvén wave packets from sudden buoyancy releases inside the core (see ref. [28]). As these waves reach the surface of the fluid core, they focus their energy towards the equatorial plane and along the strong magnetic flux lines, making the sharp interannual core flow changes. That is, the geomagnetic jerks can be associated with the acceleration of the azimuthal flow motions[28], which may associate with the significant angular momentum exchanges between the core and the mantle[6], and thus to excite the LOD variations. Meanwhile, the amplitude increasing of the 8.6-year signal is possibly induced by a three-dimensional energy-focusing mechanism[28] related to the arrivals of these localized Alfvén wave packets. Here an additional point is worthy of further discussion, i.e., if the 8.6-year signal origin is attributed to the trapped waves in a stratified layer at the Earth' core surface or the so-called QG-Alfvén waves, then the fast torsional waves detected by ref. [1] will only correspond to the 6-year oscillation. Consequently, depending on the physics chosen, the link to the magnetic field within the FOC will differ.

Based on the numerical simulations and analysis made in this work, the proposed method (i.e., NMWT+BEPME) can be used to quantitatively isolate the target harmonic (including the damping[13,46] and the increasing) signals with much high-frequency resolution, at the same time, the phase information of the target harmonic signals can also be recovered perfectly (see Supplementary Figs. 3, 9, 10, 14, 15). Hence, the 8.6-year time-domain signal recovered by this work and its fitting result (Fig. 4) provide us a strong clue for possible prediction of the future rapid geomagnetic field changes. Nevertheless, given that the geomagnetic jerks may originate from a stochastic process within the FOC[39,45], hence, it is still difficult to make an accurate prediction of the epochs of future geomagnetic jerk occurrence. Despite this, the occurrence of recent geomagnetic jerks was suggested to present an oscillatory behavior[32,35], while this work further provides a directly observed evidence to show this oscillatory behavior, which means that the jerk occurrence should not be

completely random or unpredictable. If based on the good correspondence revealed by this work, one can predict that a new geomagnetic jerk will happen (with high probability) during the period of 2020–2021.

## Methods

**NMWT method.** Defining the time signal as $h(t) \in L^1(R)$, here

$$L^1(R) = \left\{ h(t) \middle| \middle| \int_{-l}^{+l} h(t)\mathrm{d}t \middle| < +\infty, \forall l \in R^+ \right\}$$

The mathematical expression of the NMWT is written as following (ref. [12]):

$$W_g h(a,b) = \frac{1}{|a|} \int_{-\infty}^{+\infty} h(t)\bar{g}\left(\frac{t-b}{a}\right)\mathrm{d}t, \ a,b \in R, a \neq 0 \quad (2)$$

where, $a$ and $b$ are the scale and time translation factors, respectively, $g(t)$ is the so-called normal Morlet basis function, which differs from the traditional Morlet wavelet basis function. The expression of $g(t)$ is expressed as

$$g(t) = \frac{1}{\sqrt{2\pi}\sigma} e^{-\frac{t^2}{2\sigma^2} + i2\pi t} \quad (3)$$

where $\sigma$ is the window-width factor, which determines the frequency-resolution of the NMWT method, and $\sigma$ is larger, the corresponding frequency-resolution will be higher. Nevertheless, as to the $\sigma$ value, which is not the bigger the better, since $\sigma$ is also related to the edge effect range (see the following BEPME strategy). Furthermore, the $\sigma$ value is here adopted to be 3, which is large enough to distinguish the target intradecadal signals existing in the LOD variations.

As to a harmonic signal $h(t)$, which is expressed by $h(t) = A_0 \exp(i\omega(t - t_0))$, where $\omega = \frac{2\pi}{T}$. Here, defining the scale factor $a > 0$, there are two useful properties of NMWT in recovering the target signals as following (the proof can be seen in ref. [12]):

Property 1. $W_g h(T, b) = h(b), (\forall t = b, a = T)$

Property 2. $\frac{\partial}{\partial a}\left| W_g h(a,b) \right| = 0, (\forall a = T)$

The above properties of the NMWT method is also called as the inaction method[46,47].

**BEPME strategy.** Wavelet transformation (WT) usually owns the edge effect (EE), especially when the original series is not long enough, while the periods of the target signals are relatively long (e.g., the LOD data and the intradecadal signals), then the EE will significantly influence the result amplitudes. To accurately analyze the target harmonic signals in LOD variations on the intradecadal scales using WT method, we must consider the EE and manage to eliminate this effect. This EE range at each side of the data from the NMWT method can be estimated by[12]

$$R_g(a) = 1.643\sigma|a| $$

where $\sigma$ is the window-width factor and $a$ refers to the scale factor, while, in the NMWT method, $a = T$, here $T$ is the period of the target harmonic signal, so $a$ is also called the period factor.

A common simple approach to avoid the EE is the so-called bidirectional mirror-image-symmetric extension (BME) at the beginning and the end of the original data. In the NMWT method, we may also use this extension approach. However, if we directly adopt this traditional way, then the discontinuous points may appear at the two boundaries. If this case is not considered, the signal directly extracted by the NMWT method is not ideal. How to solve this issue? Although the EE exists in the NMWT method, the phase of the target signal recovered by the NMWT method is unbiased[12,13]. We can make full use of this property to solve the EE problem. In this paper, we propose a simple method, that is to search for the local extreme points of the target harmonic signal with the period $T$ (i.e., $a$) near the boundaries at both sides of the target signal in the NMWT real coefficient spectrum, and then making the symmetric extension at the two extreme points.

Here, for the sake of clarity, we construct the following composite signal $Y(t)$ (see Supplementary Fig. 1a) to illustrate the relevant steps

$$Y(t) = 1.5e^{-0.00084t}\cos\left(2\pi f_1 t + \frac{\pi}{2}\right) + 0.4\cos(2\pi f_2 t) + \text{noise}(t)$$

where $f_1 = 0.0138$ cpm (cycles-per-month, i.e., the 6-year period), $f_2 = 0.0111$ cpm (i.e., the 7.5-year period), the noise $(t)$ term means a significant stochastic noise signal, $t$ is set to be in the range of [1:1:686] with the time-interval 1 month, hence, the data length is 686 months.

Here, we will give the following four steps to avoid the EE and extract the target signal (taking the simulated 6-year oscillation as an example): i.e., firstly, applying NMWT to the composite signal $Y(t)$, we obtain the NMWT spectrum (Supplementary Fig. 1b, c), then, extracting the target signal along the ridge line from the NMWT spectrum, and the result is shown in Supplementary Fig. 2 (the red curve); secondly, searching for the extreme time ($t_i$ and $t_j$) at the two boundaries of the data after the above step, and then deleting the data outside the range of $t_i \leq t \leq t_j$; thirdly, making the symmetric extension of the data after the above two-step processing (see refs. [8,13]), here, the data length of the extension part at each side should be larger than the edge effect range $R_g(a)$; finally, applying

NMWT once again to the above output, then extracting the target signal from the NMWT spectrum along the ridge-line, and then the result is presented in Supplementary Fig. 3 (the red curve).

After the above four steps, the EE of the NMWT method can be eliminated to a great extent (see Supplementary Fig. 3). This approach (i.e., boundary extreme point mirror-image-symmetric extension, we call it BEPME strategy) is proved to be an effective way to avoid the EE, and it is developed by combining the phase-unbiased feature of the NMWT method in recovering the target signal with the traditional BME method. Nevertheless, it should be noted that although the BEPME strategy adopted in this work can eliminate the EE to a great extent, the derivations caused by the EE cannot be eliminated completely. We expect that there will be a more effective strategy (than our current method) to be developed, and we are making further efforts in this regard as well.

**On the SSA method.** As the previous works (e.g., refs. [47,48]) indicated, the frequency-resolution of SSA is related to the window length parameter ($L$), choosing an appropriate $L$ value is important for SSA method to analyze the actual data series (see Supplementary Note 2), which shows that the SSA method is not an ideal approach to distinguish and accurately isolate the target intradecadal components existing in the LOD variations (see Supplementary Figs. 5–7).

**On the normal mode stochastic excitation.** The mathematical expression of a normal mode stochastic excitation can be expressed by an AR-2-damped oscillator stochastic model as following:

$$\frac{\mathrm{d}^2 y(t)}{\mathrm{d}t^2} = a_1 \frac{\mathrm{d}y(t)}{\mathrm{d}t} + a_2 y(t) + E(t) \quad (4)$$

where $y(t)$ is just the target oscillation series, $E(t)$ may be a stochastic process, here the constants $a_1 < 0$ and $a_2 > 0$.

Furthermore, formula (4) can be transformed into

$$\frac{\mathrm{d}^2 y(t)}{\mathrm{d}t^2} + 2\beta \frac{\mathrm{d}y(t)}{\mathrm{d}t} + \omega_0^2 y(t) = E(t) \quad (5)$$

where $\beta = -\frac{1}{2}a_1$ represents the damping factor; $\omega_0 = \sqrt{a_2}$ expresses the damped oscillation frequency, and $\omega_0 = \frac{2\pi}{T_0}$, here $T_0$ expresses the oscillation period. In physics, formula (5) is called as the forced damped oscillation differential equation, where $E(t)$ is also called as the excitation term.

The 6-year oscillation in LOD can be attributed to the mantle–IC gravitational coupling oscillation mode under the action of the electromagnetic coupling effects[18,40], which just can be expressed by the formula (5). The analytical solution to the formula (5) can be written as the following convolution form:

$$y(t) = E(t)^* \varphi(t) = e^{-\beta t}\int_0^t E(\tau)e^{\beta\tau}\sin[\omega_0(t-\tau)]\mathrm{d}\tau \quad (6)$$

where * stands for the convolution operator, and $\varphi(t) = e^{-\beta t}\sin(\omega_0 t)$, which is named as the damped oscillation normal function, and $\beta$ is called as the damping factor (or the theoretical quality factor). The simulation tests of the AR-2-damped stochastic oscillation series are shown in Supplementary Figs. 16–19.

**Angular momentum budget from the solid IC.** Considering the gravitational coupling interaction between the mantle and the IC without involving the other coupling effects (e.g., electromagnetic coupling, viscous coupling). Assuming that the observed 6-year oscillation is attributed to the pure MICG coupling mode, the 6-year oscillation (i.e., $\Delta$LOD) is related to the axial rotation angular velocity of the IC departing from the gravitational equilibrium position[16–20]. Since the mantle and the IC consists of a gravitational coupling system, the IC may depart from the equilibrium state under the action of a random torque predicted by the geodynamo[16,19], under the condition of the angular momentum conservation, the angular momentum will transfer from the IC to the mantle to cause the corresponding LOD variations. According to the angular momentum conservation law, at any time $t$, the IC axial rotation angular velocity $u_i(t)$ and the angular velocity of the $u_m(t)$ satisfies the following relationship:

$$u_m(t) = -\frac{C_i}{C_m}u_i(t) \quad (7)$$

According to the relationship between $u_m(t)$ and the LOD variations (i.e., $\Delta$LOD)

$$u_m(t) = -\frac{2\pi}{(\text{LOD}_0)^2}\Delta\text{LOD}(t)$$

so

$$u_i(t) = \frac{2\pi}{(\text{LOD}_0)^2}\frac{C_m}{C_i}\Delta\text{LOD}(t) \quad (8)$$

When $|\Delta\text{LOD}(t)| = 0.12$ ms, we can estimate the magnitude of $u_i(t)$ is 0.22°/year. Importantly, this IC rotation rate (~0.22°/year) required by the 6-year oscillation is consistent with that inferred by the seismology, for example, seismic normal mode inferred that this rate is ±0.2°/year (ref. [49]), while the earthquake doublets indicated that this rate is 0.25~0.48°/year (ref. [50]). Nevertheless, here it

should be noted that, if no angular momentum is carried by FOC, one actually cannot explain LOD changes with angular momentum only in the IC (i.e., it is not possible to ignore the FOC in this balance). Given that the pure MICG mode corresponds to a zonal velocity of ~4.6 km/year at the IC equator, which is about 10 times what is inferred from geomagnetic field changes (e.g., ref. [1]), this is one reason for accounting for the fluid core motions in the case of a MICG mode, another reason is the electromagnetic coupling effects between the IC and the FOC, which will strongly couple the two (see ref. [40]).

## Data availability

The observed data that support the findings of this work are available from the International Earth Rotation and Reference System Service (IERS) website (https://www.iers.org/IERS/EN/Data Products/Earth Orientation Data/eop.html). The relevant simulation data are included in this manuscript and its supplementary files.

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

## Acknowledgements

We also thank Xinhao Liao, Guocheng Wang, Xueqing Xu, and Cancan Xu for their helpful discussion. We thank IERS website for providing LOD data. This work is supported by the B-type Strategic Priority Program of the Chinese Academy of Sciences (Grant No. XDB41000000) and National Natural Science Foundation of China (Grant No. 11803064, 11773058, 41774017, and 11373058).

## Author contributions

P.D. and C.H. performed the primary analysis; P.D. led the primary writing of the manuscript and C.H. also contributed to writing this manuscript.

## Competing interests

The authors declare no competing interests.
