## [Peer Review File · Nature Communications]

Reviewers' comments:

Reviewer #1 (Remarks to the Author):

This manuscript describes a two-part wavelet based method to distinguish the time varying character of intradecadal length-of-day (LOD) variations. The authors propose that LOD variations show the superposition of two periodic signals, one of which is well known, and one of which is new. The latter signal is suggested to coincide with occurrences of geomagnetic jerks and is suggested to be excited by MHD waves in the outer core, which might produce both the angular momentum changes seen in LOD and the magnetic signals of jerks.

I find this to be a well-written article, the method appears sound, and the proposed conclusions would potentially be a significant and well-cited contribution to the field. Similar work on the ~6yr period in LOD and linking this to geomagnetic jerks has been published as a Nature Letter (Holme and De Viron, 2013), and works on the dynamic origin of jerks have also been published in Nature (Bloxham et al 2002) and Nature Geoscience (Aubert and Finlay 2019).

Despite this, I do not support publication of this manuscript in the current form; I think the manuscript needs a significant revision to consider the major comments below as this would potentially alter the significance of the conclusions. I also attach an annotated manuscript and supplement with further, minor comments.

1. There is a failure to cite and discuss a range of literature that cover an important feature of geomagnetic jerks that is very relevant here. Discussion of this should be included, and the conclusions would be much strengthened if the authors could contribute to clarifying this issue. Three strands of this discussion are outlined below for clarity.

Jerks are generally accepted to be localised expressions at the Earth's surface, occasionally observed over large parts of the globe, which nevertheless do not occur at the same time in all regions of the globe, if seen at all in a given region. It is notoriously difficult to define a single "jerk time" from observations when many local secular accelerations overlap in time and space. Our ability to time jerks accurately and therefore our ability to assess correlations with other phenomena is far from certain. This is not discussed in this work and I believe it should be explicitly. It does not necessarily take away from the association of LOD and jerks to do so, but it may temper the certainty of the conclusions reached in this work.

Uncited works such as Nagao et al 2003 (JGR), Pinheiro et al 2011 (G3) and Brown et al 2013 (PEPI) discuss the issues of determining jerk times, uncertainties on those times, delays in observing jerks at the Earth's surface due to mantle conductivity and the difficulty of pinning down a single global epoch at which a jerk occurred. In these works, and others, jerks are somewhat confusingly considered as secular variation changes that manifest in <1yr in any given place, but also a single jerk "event" may also be considered to be distributed over several years when observations from multiple locations are considered. A key point with jerks is that the older literature often considered only the East component of the field as seen at ground geomagnetic observatories in their analyses (less external field contamination than other components), and therefore present an incomplete view of jerks that has retrospectively been fleshed out.

A second branch of work from authors such as the cited Chulliat (et al 2015, GRL) (and e.g. Soloviev et al 2017 (PEPI), Finlay et al 2015 (EPS)), relates to the view of jerks being defined not as observed surface secular variation expressions, but as pulses of secular acceleration at the CMB. This view also presents jerks as being uncertain in their timing. Soloviev et al 2017 (PEPI) note the uncertainty relating magnetic observations at the surface to the pulses at the CMB is likely +/-1yr due to filtering through a weakly conducting mantle, plus an uncertainty of 1-2yr on the timing of the peaks of their secular acceleration pulses.

I'd also suggest referring to Gillet's 2019 Chapter 9 in "Geomagnetism, aeronomy and space weather: a journey from the Earth's core to the sun" (Cambridge). There it is argued that timing jerks precisely is not possible, and that the apparent pulsing behaviour of the secular acceleration is a result of the filtered view we observe of a stochastic process within the core. This would negate any physical interpretation of the 6-year periodic signal seen in the magnetic field.

2. There are two references to information being provided in the SI that I do not think is either cited, or shown, adequately in the SI:

- a. The processing to derive the decadal LOD variation
- b. The inability of TMWT to resolve two periodic trends as close in frequency as 6yr and 8.6yr

Reviewer #2 (Remarks to the Author):

please find attached my report on the paper entitled "Intradecadal variations in length-of-day and their correspondence with geomagnetic jerks", submitted to Nature Communication by P. Duan and C. Huan.

Review on the paper entitled « Intradecadal variations in length-of-day and their correspondence with geomagnetic jerks », submitted to Nature Communication by P. Duan and C. Huan.

The authors present an analysis of subdecadal length-of-day (LOD) series, based on a normal Morlet wavelet transform (NMWT) method. They isolate two signals at 6 and 8.6 yrs, and propose a correspondance between extrema of the latter signal and the occurrence of geomagnetic jerks. They also discuss potentially important consequences of their analysis, regarding for instance the strength of the magnetic field deep in the core, or the predictability of jerks.

I have several major concerns about this work. First I would ask for additional synthetic tests regarding the analysis of LOD series. Second there are alternative choices possible for the catalogue of jerks, which should be considered. This requires a discussion on the current debate about the existence of jerks isolated in time. Finally, some geophysical consequences should be revised. I detail these points below, and recommend major revisions for a publication to Nature Communication.

Major remarks :

1) additional synthetic tests :

I acknowledge the efforts by the authors to perform synthetic tests. Nevertheless, I wonder if the two spectral lines found with the wavelet analysis could still be the consequence of too short series. I raise this question for two reasons. First in Fig. 1b, there seem to be some drift (in period) of the maximum of energy between 6 and 8.6 yr (with some signal around 7 yr too !). Second, a modulated monochromatic signal naturally tends to show, if analysed on a short series, several spectral bands. This is particularly tricky since the LOD spectrum is red. Long periods may pollute the analysis : this effect is considered in the supplementary material, but the synthetic trend considered there (saw-toothed) might be mis-leading. It indeed contains strong hypotheses, as spectral properties of such series is specific and well distinct from that of sine curves.

I thus think the authors should make the following test :

- generate a AR-2 damped oscillator stochastic series (choosing first a single period at 6 yr, and a realistic quality factor) ;
- add-it to an AR-1 stochastic process of significantly larger r.m.s. amplitude (e.g. x5), with a decay time long compared with 6 yr ;
- apply their method on series as long as the available LOD series... is there a single or two spectral lines that show up ? If more than one, the authors results are doubtful, otherwise the test is not negative.

This way, the author could furthermore test if they recover the decay rate associated with the « true » damped oscillator series. Finally the same test could be performed by considering two damped oscillators instead of one.

I also ask these questions since SSA on LOD series cleaned from atmospheric winds and solid tides contributions clearly show a 6 yr line, but not the one at 8.5 yr.

Finally, why is this new line not apparent in previous studies implying the authors [8,13] ? is it only because the spectra were cut in Figures at periods shorter than 8 yrs ?

2) jerks catalogues :

The authors refer to jerks dated from [6, 23, 24]. There exists two systematic studies of such events that the authors shall consider : Brown et al (2013) and Soloviev et al (2017). In the former, more frequent jerks are found as data become more accurate, almost filling the entire span at recent epochs. In the second, SA pulses (in between jerks) are found at preferred epochs, with consequences on the dates of jerks (some being less v-shaped than others). Jerks dates from these two studies should be considered when building and discussing Fig. 4.

Furthermore, there is currently a strong debate on the existence of jerks localized in time, which should be recalled in this study. The localization of jerks in time could be the effect of considering filtered series (because we cannot separate well internal from external sources). It has some implications on the spectra of SV series. Indeed, SA pulses seem to occur at preferred epochs (see

Soloviev et al), but SV series do not show clear isolated spectral lines at subdecadal periods – see for instance to Gillet (2019) and to Lesur et al (2017). The sentence « the above phenomenon... » on p. 6 should be modified accordingly.

3) geophysical consequences :

- The discussion at the end of p. 6 should be strongly rephrased. I do not see why « Fig. 4 also reveals that the increasing... these jerks ». This whole paragraph seems highly speculative.

- The authors claim (p. 7) that the 6 yr LOD changes is « most likely » excited by an inner core swing. This sentence is not correct. This is one possibility among others. The inner core is only responsible for a tiny fraction of the angular momentum budget, and the LOD series alone cannot be considered as a proof of inner core oscillations. On the other hand, angular momentum associated with TW alone explains subdecadal LOD changes, and there is yet no definitive scenario for their triggering. It may be associated with a swing of the inner core, but alternatively Lorentz torques on the inner core, or within the bulk, could equally well generate waves travelling from the inner core (see Teed et al 2014 ; Gillet et al, 2017).

- The authors claim (p. 8) that a period of 8.6 yrs for torsional waves (TW), instead of 6 yrs, should lead to revise down the strength of the magnetic field deep in the core. This is without considering that the period is not immediately related to the field intensity. Actually this latter is constrained by the slowness (integral, along the cylindrical radius, of the inverse of the Alven speed, see Gillet et al, 2017). This latter defines the waveform, and it is the waveform that leads to the estimate of the field intensity in the core by [1]. As a consequence, one cannot follow the argument on a lower field intensity.

- Next the authors mention (still on p. 8) that waves in a stratified layer have been « detected » ; this is not correct. What Chulliat et al (2015) have shown is that one may possibly interpret subdecadal SV changes as the signature of such waves. This paragraph is speculative, especially since in such a case, associated surface core flows would not explain directly LOD changes anymore – see also the recent work by Gastine et al (2019) that does not favor the presence of a stratified layer at the top of the outer core.

- Alternatively, the authors should discuss further the work by Aubert & Finlay (2019), in particular the link these authors see between jerks and the acceleration of zonal motions (and thus possible links to LOD variations, as in the study by Holme & de Viron, 2013).

- Finally, the authors mention the « obvious » periodicity in jerk occurrence (see p. 9). This statement should be softened (see above point 2). They also speculate on the possibility to predict jerks. This idea should be further developed in the light of the results by Aubert & Finlay (2019) ; also a discussion of the analysis of dynamo simulations by Gillet et al (2019) would be welcome. These authors find that a large fraction of rapid SV changes are due to unmodelled SV associated with subgrid-scale induction... how predictable are these, if not correlated with large length-scale dynamics ?

Minor points :

- p. 2 : the last sentence seems rather speculative at this stage. It should be removed. There is no definitive evidence of a triggering of TW by oscillations of the inner core.

- p. 6 : why refer to [22] here ? There exists more accurate references.

- p. 8 : why refer to [29] for the excitation of torsional waves on the inner core ? There are better references (e.g. Mound & Buffett, 2006).

- method section : isn't it an ambiguity when extending with the symmetric series in a case where there are 2 sine functions ? I guess that if one is larger than the other it may work, but what happens otherwise ?

references :

- Aubert & Finlay, Geomagnetic jerks and rapid hydromagnetic waves focusing at Earth's core surface, *Nature Geoscience* 12, 393-398, 2019

- Brown et al, Jerks abound: An analysis of geomagnetic observatory data from 1957 to 2008, *Phys. Earth Planet. Int.* 223, 62-76, 2013
- Chulliat et al, Fast equatorial waves propagating at the top of the Earth's core. *Geophys. Res. Lett.* 42(9), 3321-3329, 2015
- Gastine et al, Dynamo-based limit to the extent of a stable layer atop Earth's core, *arXiv preprint arXiv:1910.05102*, 2019
- Gillet et al, Excitation of travelling torsional normal modes in an Earth's core model, *Geophys. J. Int.* 210(3), 1503-1516, 2017
- Gillet, Spatial and temporal changes of the geomagnetic field: insights from forward and inverse core field models, in *Geomagnetism, Aeronomy and Space Weather: a Journey from the Earth's Core to the Sun*, 2019
- Gillet et al, A reduced stochastic model of core surface dynamics based on geodynamo simulations. *Geophys. J. Int.* 219(1), 522-539, 2019
- Holme & de Viron, Characterization and implications of intradecadal variations in length of day, *Nature* 499, 202–204, 2013
- Lesur et al, On the frequency spectra of the core magnetic field Gauss coefficients, *Phys. Earth Planet. Int.* 276, 145-158, 2018
- Teed et al, The dynamics and excitation of torsional waves in geodynamo simulations, *Geophys. J. Int.* 196, 724–735, 2014
- Soloviev et al, Detection of secular acceleration pulses from magnetic observatory data, *Phys. Earth Planet. Int.* 270, 128-142, 2017

Reviewer #3 (Remarks to the Author):

Review of "Intradedecadal variations in LOD...." By Duan and Huang

Reviewer: Richard Holme

Recommendation: Not acceptable in current form

Note: this review is phrased in a rather personal way. This is not because I am upset that the authors are trying to supersede my work – more power to them! – but because what they are doing is antithetical to my approach – I seek simplicity, without relying on a *duex ex machina* from complicated maths – and I would argue that my methods achieve a better fit to data. Fundamentally, which is correct may not be answerable, although I suggest a test below.

This paper examines the variation in LOD. It is a development of an earlier paper (by me!) which argued that intradedecadal variation in LOD could be modelled almost entirely by a 5.8 year period signal of constant amplitude. Such a model was not a perfect fit – there were clearly jumps in the LOD which can not be fit by a simple harmonic signal – but nonetheless, it explained most of the variance. One additional point in that paper is relevant to this work – the longer period decadal variation was removed by a smooth curve fit, and this fit was repeated iteratively after subtracting the harmonic variation – the variation requires detrending with the long period signal, while the long period signal required detrending the oscillation before it is fit.

This paper instead uses various rather more developed techniques (Morlet wavelets, BEPME (which I will not write in long form!)) to attempt to separate the longer period signal, then shows a good fit to the residual with wavelet methods, and then fits the wavelet signals with two different harmonic signals – one of about 6 year period and 8.6 year signal, the former decaying, the latter growing. The authors argue that this model better fits the data.

First I must say that this problem is strongly non-unique – both the earlier formalism and the work in this paper are possible fits to the data, and on some level it is not possible to judge which of the two is "better". Where the two methods differ is the differing reliance on complex methods. This paper relies on its methodology, and in particular a spectral separation – the background signal is obtained by a low pass filter, defining it in terms of its period. This would work if the signal was longer (so more than a few periods) or genuinely appeared stationary, but it does not, either at long periods or short periods. The remaining shorter periods are then fit very well by the wavelet transform – but this is inevitable, as the wavelets have in effect almost a continuum of free parameters. The final argument for the growing and decaying exponentials, leading to a 6 parameter fit – two amplitudes two phases, two decay rates) does not fit the data particularly closely (see figure 3, and compare with figure S3 from my earlier paper for a fit with only a 2 parameter system).

What is the necessity of the two different periods? This relates to the removal of a slowly varying trend. This trend has sufficient variation that it can covary with the harmonic components solved for here. Thus the problem, even as established, is seriously nonunique. In the original paper, I iterated between the long-term trend and the oscillation – I removed the oscillation before detrending again, and by doing this got a much better fit for a single constant amplitude component. This is not done here – the problem can be seen in 1992 where the background trend shows a peak at exactly the same time as the 6 year oscillation. The iterative approach allowed this variation to be adjusted by the background trend, the approach in this paper requires a low value of the oscillation at this time, which is where the mix of the 6 year and 8 year periods comes in – at this point they are out of phase, and so the background trend does not need to be altered. To avoid this cancellation happening at other

times, the growing and decaying exponentials mean that there is not cancellation apart from at this one time.

Both papers fit the data, and therefore on some level it is not possible to argue that one is better than the other. However, the simplicity of the first paper (a smooth curve combined with a two parameter fit) compared with complexity of the second (highly developed spectral methods for which the time series may be too short, with a 6 parameter fit that fits the data less well) suggests to my mind that this work does not provide an advance in our understanding. Note also that the fit is not perfect in either case – this is because the signal being fit is not stationary, and shows jumps (probably due to physical processes) – again see figure S3. So having arguments about including higher complexity of statistically stationary elements is a bit futile. This argument would need a discussion of how great an improvement in fit the more complex model provides, including allowing time variation of the background signal – and I would argue that instead the fit is worse.

The big problem is the apparent belief that complicated modelling methods (such as wavelets) will solve this problem. Even simple analysis of splitting different frequencies of signals is not robust – it is highly likely that there are components of intervening frequencies in both the background signal and the oscillation. The methods rely on a (non-stationary) series of (noisy) data, with insufficient length to justify complex analysis – and yet the final fit is not as good as the simpler analysis. There is one possible test – I note in figure 3, the data end at 2015, so there should be another 4 years of data by now to provide a test of just how good a predictive model is provided by the two models.

The discussion of the physical mechanisms does not bring anything new, or anything robust. The problem with such a speculative analysis, as I have found to my cost, is that even if qualified (not really done here), once it enters the literature, other workers seem to believe it proven and then chase off in unjustified directions of work. There is reference to work of Gilet – again this relies on the validity of the transform methods and a clear separation between short and long period signal, so at least it is good that the two analyses are in agreement. However, I note that the flow analysis that Gilet performs ended up with just a 6 year variation – in fairness, long before I got involved in this story.

Minor points –

Reference 6 talks about intradecadal variation, but does not identify the 6 year period as clearly separated. This reference needs to appear, but not at the point in the text where it now does.

What AAM series is used? Is there any smoothing of the data after this removal (a running average or a low pass filter?)

Response to reviewers

Reviewer #1 (Remarks to the Author)

This manuscript describes a two-part wavelet based method to distinguish the time varying character of intradecadal length-of-day (LOD) variations. The authors propose that LOD variations show the superposition of two periodic signals, one of which is well known, and one of which is new. The latter signal is suggested to coincide with occurrences of geomagnetic jerks and is suggested to be excited by MHD waves in the outer core, which might produce both the angular momentum changes seen in LOD and the magnetic signals of jerks.

I find this to be a well-written article, the method appears sound, and the proposed conclusions would potentially be a significant and well-cited contribution to the field. Similar work on the ~6yr period in LOD and linking this to geomagnetic jerks has been published as a Nature Letter (Holme and De Viron, 2013), and works on the dynamic origin of jerks have also been published in Nature (Bloxham et al 2002) and Nature Geoscience (Aubert and Finlay 2019). Despite this, I do not support publication of this manuscript in the current form; I think the manuscript needs a significant revision to consider the major comments below as this would potentially alter the significance of the conclusions. I also attach an annotated manuscript and supplement with further, minor comments.

Response: Thank the reviewer very much for the valuable comments and the good evaluation on our manuscript. According to these comments and suggestions, we have further revised the manuscript and made the relevant discussion in the revised manuscript. We believe that these suggestions can well improve the quality of our work and may play an important role in guiding our future work.

1. There is a failure to cite and discuss a range of literature that cover an important feature of geomagnetic jerks that is very relevant here. Discussion of this should be included, and the conclusions would be much strengthened if the authors could contribute to clarifying this issue. Three strands of this discussion are outlined below for clarity.

Response: Thank the reviewer very much for the good suggestions. The initial idea and the

original thoughts of this work were: 1) Detecting the intradecadal variations in LOD and reporting an 8.6-year component amplitude increasing phenomenon; 2) Reporting a very good correspondence relationship between the extremes of the 8.6-year signal and the jerk occurrence times. In the original manuscript, although we referred to the relevant relationship between the jerks and the 8.6yr oscillation in LOD, we did not further discuss the inherent physical links between them, while we did not give the introduction of the jerk occurrence features, timing uncertainties, and the relevant debates from the different published works. Here, thank the reviewer very much for proposing the valuable and constructive suggestions, the related arguments and their related references.

According to these suggestions, we further add the relevant discussion and references to the revised manuscript, and we have also illustrated the jerk definition and the jerk important features. Please the revised manuscript, for example, the blue fonts in page 7, lines 157th~177th.

Jerks are generally accepted to be localised expressions at the Earth's surface, occasionally observed over large parts of the globe, which nevertheless do not occur at the same time in all regions of the globe, if seen at all in a given region. It is notoriously difficult to define a single "jerk time" from observations when many local secular accelerations overlap in time and space. Our ability to time jerks accurately and therefore our ability to assess correlations with other phenomena is far from certain. This is not discussed in this work and I believe it should be explicitly. It does not necessarily take away from the association of LOD and jerks to do so, but it may temper the certainty of the conclusions reached in this work.

Uncited works such as Nagao et al 2003 (JGR), Pinheiro et al 2011 (G3) and Brown et al 2013 (PEPI) discuss the issues of determining jerk times, uncertainties on those times, delays in observing jerks at the Earth's surface due to mantle conductivity and the difficulty of pinning down a single global epoch at which a jerk occurred. In these works, and others, jerks are somewhat confusingly considered as secular variation changes that manifest in <1yr in any given place, but also a single jerk "event" may also be considered to be distributed over several years when observations from multiple locations are considered. A key point with jerks is that the older literature often considered only the East component of the field as seen at ground geomagnetic observatories in their analyses (less external field contamination than other components), and

therefore present an incomplete view of jerks that has retrospectively been fleshed out.

A second branch of work from authors such as the cited Chulliat (et al 2015, GRL) (and e.g. Soloviev et al 2017 (PEPI), Finlay et al 2015 (EPS)), relates to the view of jerks being defined not as observed surface secular variation expressions, but as pulses of secular acceleration at the CMB. This view also presents jerks as being uncertain in their timing. Soloviev et al 2017 (PEPI) note the uncertainty relating magnetic observations at the surface to the pulses at the CMB is likely +/-1yr due to filtering through a weakly conducting mantle, plus an uncertainty of 1-2yr on the timing of the peaks of their secular acceleration pulses.

Response: Thank the reviewer very much for the valuable and constructive suggestions. As the reviewer suggested, jerks are generally localized expressions at the Earth's surface, occasionally observed over large parts of the globe, which nevertheless do not occur at the same time in all regions of the globe due to mantle conductivity (e.g., Nagao et al, 2003; Pinheiro et al, 2011), we have added this discussion in the revised version, please see the revised manuscript, i.e., page 7, lines 171th~174th.

This is a nice suggestion that it is far from certain to time jerks accurately, and therefore, to assess correlations with other phenomena, while this work just attempts to isolate a new significant harmonic component (i.e., the 8.6yr signal) with an increasing amplitude from the LOD intradecadal variations using the normal Morlet wavelet transformation (NMWT) with a high-frequency resolution, and furthermore, combining with the previous related published works, we try to discuss the physical origin of the 8.6yr oscillation and explore its inherent physical links to the geomagnetic jerks. Therefore, this work is just an effort in exploring the relationship between jerks and “other physical phenomena” mentioned by the reviewer. According to the previous works (e.g., Holme and de Viron, 2013; Chulliat and Maus, 2014; Torta et al, 2015), we can give the following 11 jerk events during 1962~2018, (i.e., 1969, 1972, 1978, 1982, 1986, 1991, 1999, 2003.5, 2007, 2011 and 2014). Of course, as mentioned above, these listed 11 jerk timings may be not accurate enough, as the reviewer suggested, it is difficult to define a single “jerk time” from observations when many local secular accelerations overlap in time and space. However, the above jerk timings were suggested to be the best determinations (see Holme and de Virion, 2013), meanwhile, this work shows that there is a very good correspondence relationship between these timings with the extremes of the 8.6yr oscillation (see Figure 4 in this manuscript), for example,

Pinheiro et al (2011) specialized studied the following four most well-known jerks (i.e., 1969, 1978, 1991, 1999), all of which well correspond to the extremes of the 8.6yr oscillation recovered by this work, which, in our opinion, is a very interesting phenomenon. In summary, we hope that this work can take an important step towards (or provide an entry for) finally solving this "certain" problem provided by the reviewer.

According to the above suggestions, we have also further cited the works (Nagao et al, 2003; Pinheiro et al, 2011; Brown et al, 2013; Soloviev et al, 2017 and Finlay et al, 2015) and further made the relevant discussion in the revised manuscript. In our original manuscript, we found that the periods during these listed jerk timings (from the previous works) and thereafter less than 1yr always corresponds to the extremes of the 8.6yr oscillation, thus we called this 1yr as the duration of the jerk event. However, considering the above comments provided by the reviewer and the conception of the jerk occurrence epoch and the duration definition from the previous works (e.g., Nagao et al 2003), it is not appropriate to call the 1-year as the jerk duration, we have revised this point in the revised manuscript, please see the revised version, page 7, lines 167th~177th.

In addition, we agree with the viewpoint that the older literature often considered only the East component of the field as seen at ground geomagnetic observatories in their analyses (less external field contamination than other components), and therefore present an incomplete view of jerks that has retrospectively been fleshed out.

In summary, the reviewer provided some constructive suggestions for our work. As the reviewer suggested, the jerks being defined as pulses of secular acceleration at the CMB from the previous works (e.g., Soloviev et al 2017, Finlay et al 2015) also shows jerks as being uncertain in their timing. In summary, according to the good suggestions and the above discussion, we have further considered these points and discussed them in the revised manuscript, please see the revised version, i.e., page 7, lines 160th~176th; page11, 265th~271th.

I'd also suggest referring to Gillet's 2019 Chapter 9 in "Geomagnetism, aeronomy and space weather: a journey from the Earth's core to the sun" (Cambridge). There it is argued that timing jerks precisely is not possible, and that the apparent pulsing behaviour of the secular acceleration is a result of the filtered view we observe of a stochastic process within the core. This would negate any physical interpretation of the 6-year periodic signal seen in the magnetic field.

Response: Thank the reviewer very much for providing the good suggestions. According to this suggestion, we have cited the works of (Gillet 2019; Gillet et al, 2019) in the revised manuscript and discussed the issue about the jerk stochastic occurrence, furthermore, we have added the relevant discussion to the revised manuscript, please see the revised manuscript, pages 11, lines 261th~265th.

Additionally, on the basis of the current publications (e.g., Nagao et al 2003; Holme and de Viron, 2013; Pinheiro et al 2011; and Brown et al 2013), this work further discusses the uncertainties of the jerk epochs and the potential inherent physical links between the 8.6yr oscillation and the jerks. As mentioned above, we agree with that the jerk occurrences may own a certain of randomness and it may be difficult to predict the jerk timings precisely, however, the jerks were found to be caused by the arrivals of the waves in a current simulation work (Aubert and Finlay, 2019), and these arrivals can strengthen the acceleration of the zonal flow motions to cause the corresponding LOD changes, furthermore, previous works (e.g., Torta et al, 2014; Kotzé, 2017) showed that the jerk occurrence presents 3~5yr periodicity over the last several decades, and moreover, this work provides the observed evidence of the correspondence relationship between jerks and the 8.6yr oscillation during the past several decades. These points show that the jerk occurrence is very likely to be not completely random or unpredictable. As Aubert and Finlay (2019) indicated, jerks are the major obstacle to predict the geomagnetic changes ahead, therefore, this work just attempts to find a new way to predict the jerks ahead through the 8.6yr oscillation in LOD, while this paper attempts to provide a new possible physical links between the jerk occurrence and the 8.6yr oscillation in LOD, which is also an effort to predict the occurrence of jerk in advance through the 8.6yr time-domain oscillation recovered by this work.

In summary, in the revised manuscript, we have further cited the previous works (e.g., Gillet et al, 2017; Gillet et al, 2019; Gillet 2019) on the assumption of a stochastic process within the core and discussed them. According to the above discussion and the reviewer's suggestions, we have further revised the manuscript, page 8, lines, 190th~194th; page 11, lines 265th~271th; pages 11-12, lines 289th~293th.

2. There are two references to information being provided in the SI that I do not think is either cited, or shown, adequately in the SI:

- a. The processing to derive the decadal LOD variation
- b. The inability of TMWT to resolve two periodic trends as close in frequency as 6yr and 8.6yr

Response: Thank the reviewer very much for the good suggestions. According to these suggestions, we have further revised the supplementary information (SI) of this work and further provided the relevant references in the SI. Please the supplementary reference part of the revised SI, i.e., page 23~24, lines 440th~477th.

References

- Aubert, J., Finlay, C.C. Geomagnetic jerks and rapid hydromagnetic waves focusing at Earth's core surface. *Nature Geoscience*, 12, 393-398 (2019).
- Brown, W.J., Mound, J.E., Livermore, P.W. Jerks abound: An analysis of geomagnetic observatory data from 1957 to 2008. *Phys. Earth Planet Inter*, 233, 62-76 (2013).
- Chulliat, A., Maus, S. Geomagnetic secular acceleration, jerks, and localized standing waves at the core surface from 2000 to 2010. *J. Geophys. Res: Solid Earth*, 119, 1531-1543 (2014).
- Finlay, C., Olsen, N., Tøffner-Clausen, L. DTU candidate models for IGRF-12 and the CHAOS-5 geomagnetic field model. *Earth Planets Space*. 67:114, doi:10.1186/s40623-015-0274-3 (2015)
- Gillet, N., Jault, D., Canet, E. Excitation of travelling torsional normal modes in an Earth's core model. *Geophys. J. Int.* 210, 1503-1516 (2017).
- Gillet, N., Huder, L., Aubert, J. A reduced stochastic model of core surface dynamics based on geodynamo simulations. *Geophys. J. Int.* 219(1), 522-539 (2019).
- Gillet, N. Spatial and temporal changes of the geomagnetic field: insights from forward and inverse core field models. *Geomagnetism, Aeronomy and Space Weather: a Journey from the Earth's Core to the Sun* (2019).
- Holme, R., de Viron, O. Characterization and implications of intradecadal variations in length of day. *Nature*. 499, 202-205 (2013).
- Kotzé, P.B. The 2014 geomagnetic jerk as observed by southern African magnetic observations. *Earth Planets Space*. 69:17, doi:10.1186/s40623-017-0605-7 (2017).
- Nagao, H., Iyemori, T., Higuchi, T., Araki, T. Lower mantle conductivity anomalies estimated from geomagnetic jerks. *J. Geophys. Res.* 108 (B5), 2254, doi:10.1029/2002JB001786 (2003).
- Pinheiro, K.J., Jackson, A., Finlay, C.C. Measurements and uncertainties of the occurrence time of the

1969,1978,1991,and 1999 geomagnetic jerks. *Geochem. Geophys. Geosyst.*,12,Q10015, doi:10.1029/2011GC003706 (2011).

Soloviev, A., Chulliat, A., Bogoutdinov, S. Detection of secular acceleration pulses from magnetic observatory data. *Phys. Earth Planet. Inter.*, 270:128-142 (2017).

Torta, J.M., F. J. Pavón-Carrasco, S. Marsal, and C. C. Finlay. Evidence for a new geomagnetic jerk in 2014, *Geophys.Res. Lett.*, 42, 7933–7940, doi:10.1002/2015GL065501 (2015).

Reviewer #2 (Remarks to the Author)

Review on the paper entitled « Intradecadal variations in length-of-day and their correspondence with geomagnetic jerks », submitted to Nature Communication by P. Duan and C. Huang.

The authors present an analysis of subdecadal length-of-day (LOD) series, based on a normal Morlet wavelet transform (NMWT) method. They isolate two signals at 6 and 8.6 yrs, and propose a correspondance between extrema of the latter signal and the occurrence of geomagnetic jerks. They also discuss potentially important consequences of their analysis, regarding for instance the strength of the magnetic field deep in the core, or the predictability of jerks. I have several major concerns about this work. First I would ask for additional synthetic tests regarding the analysis of LOD series. Second there are alternative choices possible for the catalogue of jerks, which should be considered. This requires a discussion on the current debate about the existence of jerks isolated in time. Finally, some geophysical consequences should be revised. I detail these points below, and recommend major revisions for a publication to Nature Communication.

Response: Thank the reviewer very much for the valuable comments and suggestions. First, we have further made the test analysis according to these suggestions; second, we have also further considered the alternative choices possible for the jerk catalogue and made the relevant discussion on the current debate about the existence of jerks isolated in time and revised the manuscript based on the suggestions. Finally, we have further discussed and revised some consequence of the geophysical results. Please see the revised manuscript of this work.

Major remarks :

1) additional synthetic tests :

I acknowledge the efforts by the authors to perform synthetic tests. Nevertheless, I wonder if the two spectral lines found with the wavelet analysis could still be the consequence of too short series. I raise this question for two reasons. First in Fig. 1b, there seem to be some drift (in period) of the maximum of energy between 6 and 8.6 yr (with some signal around 7 yr too !).

Response: Thank the reviewer very much for the good comments and questions. The two spectral lines presented in the normal Morlet wavelet transformation (NMWT) spectrum of this work are

not the consequence of the too short series, but these two obvious spectral lines that represent two harmonic components (i.e., one is a 6yr term and the other one is an 8.6yr term) indeed exist on the intradecadal scales (i.e., 5~10yr band) of the length-of-day (LOD) variations. Since the NMWT method owns the high frequency resolution (e.g., Liu et al, 2007), and this method not only can distinguish the 6yr component and the 8.6yr component, but also it can clarify the two periodic signals (i.e., 6yr and 7.2yr) with more closer frequencies. According to the suggestions provided by the reviewer, here, we will adopt two typical simulation examples to demonstrate this point.

1) Simulation of the superposition of 6yr term and 8.6yr term

The composite signal $Y(t)$ is written as following

$$Y(t)=y_1(t)+y_2(t)+noise(t)$$

where, $t=[0:599]$, the sampling rate is set to be 1 month, here the data length is 600 months (i.e.,50yrs), which is shorter than the actual length (~57yrs) of the LOD data from 1962 to 2019; $y_1(t)$ and $y_2(t)$ are the simulated 6yr and 8.6yr steady harmonic signals respectively, and their expressions are written as following

$$y_1=0.12\sin(2\pi f_1t), \quad y_2=0.08\sin(2\pi f_2t)$$

where, $f_1=0.0138\text{cpm}(\text{cycle-per-month})$, $f_2=0.00969\text{cpm}$, $noise(t)$ is a significantly stochastic white noise signal.

Furthermore, the composite signal $Y(t)$ is shown in the Figure 1, and the corresponding NMWT time-frequency spectrum of $Y(t)$ are presented in Figure 2(a) and (b). Figure 2 clearly shows that the 6yr and 8.6yr terms can be clarified by NMWT spectrum. In the next, we will further simulate three periodic components (i.e., 6yr, 7.2yr and 8.6yr) to illustrate the relevant issues.

Figure 1 The simulated composite signal $-Y(t)$

Figure 2 (a) and (b) show NMWT time-frequency spectrum. The two periodic components existing in the $Y(t)$ can be separated, and the white noise cannot influence the spectrum quality

2) Simulation of the superposition of three terms (i.e., 6yr, 7.2yr and 8.6yr)

The composite signal $Y(t)$ is written as

$$Y(t)=y_1(t)+y_2(t)+y_3(t)+noise(t)$$

Here, $y_3=0.05\sin(2\pi f_3 t)$, $f_3=0.01157\text{cpm}$, which corresponds to the 7.2yr signal.

Furthermore, we consider the following two cases:

- (1) Considering a relative smaller white noise term, and the result is shown in Figure 3(a);
- (2) Considering a relative larger white noise term (here, the above noise $\times 5$), and the result is shown in Figure 3(b);

shown in Figure 3(b);

The corresponding NMWT spectrum of them are displayed in Figure 4.

Figure 3 Simulated composite signal $Y(t)$ with different white noises

Figure 4 (a)-(b) show the NMWT time-frequency spectrum of the composite signal - $Y(t)$ with a smaller white noise; (c)-(d) displays the NMWT time-frequency spectrum of the composite signal - $Y(t)$ with a larger white noise; these two cases of NMWT spectrum are almost the same each other, which shows that NMWT owns a good robustness.

Figure 4 shows that NMWT spectrum can distinguish the three different frequency components, namely, 6-year, 7.2-year and 8.6-year components. In this work, we do not carefully focus on the “drift phenomenon” of the weak “quasi-7.2yr period” component (see Fig.1 c,d in the revised manuscript). Nevertheless, the above simulation analysis shows that this “drift phenomenon” is unrelated to the shorter data of the actual LOD series, since this simulation data length is shorter than that of the actual LOD data. Because this work mainly focuses on the 8.6yr signal in LOD and its relation to the geomagnetic jerks, and the “quasi-7.2yr periodic component” in LOD is relatively weaker, so in this work, we will not discuss this “quasi-7.2yr signal” too much, nevertheless, it may reflect the signature of the fluid outer core motions (e.g., Gillet et al, 2015; Gillet et al, 2017), in addition, we do not exclude a possibility that it may be due to the consequence of the stochastic excitation from two free normal modes (i.e., the 6yr and 8.6yr signals) (e.g., see the “case 9” in Figure 12 of this response letter), of course, the more definitive conclusion still needs to be further explored later.

In summary, the above simulation analysis shows that NMWT with the high frequency resolution can distinguish the different periodic signals (e.g., 6yr and 8.6yr) existing in the intradecadal variations of the LOD data. The two obvious spectral lines (i.e., 6yr and 8.6yr)

presented in the NMWT spectrum are not the consequences of the short series, but LOD variations own the two different frequency components.

Second, a modulated monochromatic signal naturally tends to show, if analysed on a short series, several spectral bands. This is particularly tricky since the LOD spectrum is red. Long periods may pollute the analysis: this effect is considered in the supplementary material, but the synthetic trend considered there (saw toothed) might be mis-leading. It indeed contains strong hypotheses, as spectral properties of such series is specific and well distinct from that of sine curves. I thus think the authors should make the following test :

- generate a AR-2 damped oscillator stochastic series (choosing first a single period at 6 yr, and a realistic quality factor) ;
- add-it to an AR-1 stochastic process of significantly larger r.m.s. amplitude (e.g. x5), with a decay time long compared with 6 yr ;
- apply their method on series as long as the available LOD series... is there a single or two spectral lines that show up ? If more than one, the authors results are doubtful, otherwise the test is not negative.

Response: Thank the reviewer very much for providing the good and constructive suggestions. According to these suggestions, we will do our best to answer the suggestions regarding our understanding of the stochastic excitation oscillations. If a monochromatic physical signal itself presents a modulation phenomenon in its amplitude changes, then this physical modulated signal should be the result of excitation. Therefore, according to the reviewer's suggestions, we raise the following question: If the NMWT method is used to analyze this excited monochromatic signal, then how it will perform in the NMWT time-frequency spectrum? Here, we make the relevant tests based on the suggestions from the reviewer to answer this question.

First, the mathematical expression of an AR-2 damped oscillator model is written as

$$\frac{d^2y(t)}{dt^2} = a_1 \frac{dy(t)}{dt} + a_2 y(t) + E(t) \quad (1)$$

where, $y(t)$ is just the target oscillation series, $E(t)$ may be a stochastic process, and here the constants $a_1 > 0$ and $a_2 > 0$.

Furthermore, formula (1) can be transformed into

$$\frac{d^2 y(t)}{dt^2} + 2\beta \frac{dy(t)}{dt} + \omega_0^2 y(t) = E(t) \quad (2)$$

where, $\beta = -\frac{1}{2}a_1$ represents the damping factor; $\omega_0 = \sqrt{a_2}$ expresses the damped oscillation frequency. In physics, formula (2) is called as the forced damped oscillation differential equation, where $E(t)$ is also called as the excitation term.

As to the 6yr oscillation case (i.e., $\omega_0 = \frac{2\pi}{T_0}$, here $T_0=6\text{yr}$). Duan et al (2018) indicated that

this 6yr oscillation in LOD can be attributed to the mantle-inner core gravitational coupling oscillation under the action of the electromagnetic coupling effects, which is just the same mathematical form as the formula (2). The analytical solution to the formula (2) can be expressed by the following convolution form

$$y(t) = E(t) * \varphi(t) = e^{-\beta t} \int_0^t E(\tau) e^{\beta \tau} \sin[\omega_0(t - \tau)] d\tau \quad (3)$$

where, “*” stands for the convolution operator, and $\varphi(t) = e^{-\beta t} \sin(\omega_0 t)$, which is named as the damped oscillation normal function, and β is the damping factor.

Formula (3) indicates that the time-varying characteristic of $y(t)$ essentially depends on the specific form of $E(t)$. If $E(t)$ is a stochastic white noise series, see Figure 5(a), then we can generate a forced damped oscillation series on the basis of formula (3) (here $\beta=8.4 \times 10^{-4}/\text{month}$ in Figure 5(b)), the result of which is shown in Figure 5(c). Figure 5(c) shows that the 6yr oscillation presents a modulated amplitude changes under the action of a continuously stochastic excitation. Furthermore, we apply the NMWT method to the damped 6yr stochastic oscillation series, and the result is displayed in Figure 6. Figure 6 indicates that there is still only one obvious 6yr periodic component existing in the NMWT spectrum.

Figure 5 (a) shows a simulated Gauss “white noise” stochastic series, which can be see as the “input” series (i.e., $E(t)$) of this convolution operation system expressed by formula (3); (b) displays the damped oscillation function, here $\beta=8.4\times 10^{-4}/\text{month}$; (c) shows the “output” series of this convolution operation system, which expresses the 6yr forced damped oscillation series, and this “output” series depends on the “input” series - $E(t)$.

Figure 6 The NMWT time-frequency spectrum of the “output” series

In this test, we repeatedly use different random excitation sequences $E(t)$ as the “input” series to simulate a variety of “output” results, see Figure 7. Based on our repeated simulation tests, there exists a possibility that, in some cases, the “peak splitting” phenomena appear in the NMWT spectrum, e.g., the “output 6” and “output 9” shown in Figure 7, which may be just the cases that “a modulated monochromatic signal naturally tends to show several spectral bands” suggested by the reviewer. However, we find that even though a monochromatic signal (e.g., the 6yr oscillation normal mode signal in LOD) presents a modulated phenomenon under the action of an arbitrary continuous stochastic excitation and its time-frequency spectrum may perform the similar

behaviors as the case of “output 6” presented in Figure 7, but the energy and the width of the spectral bands of these “splitting” components in NMWT spectrum are small and narrow. That is to say, the significant 8.6yr signal found by this work is not possible to be generated by this case, but we do not exclude the possibility that the weaker quasi-7.2yr signal around the 6yr period may be the consequence of this case.

According to the good comments and the above discussion, we further add the AR-2 model and the relevant simulation tests in the revised manuscript, please see the revised version, i.e., pages 14-15, lines 368th~384th; and, in supplementary information, the pages 15~19, lines 321th~362th.

Figure 7 The left columns show the various “output” results corresponding to the various “input” series, the middle and the right columns show the NMWT time-frequency spectrum of the “output” results, where the horizontal ordinates represent the time/month, while the vertical coordinates express the period/year.

In addition, in the “supplementary information” of this manuscript, we used a “saw-toothed” shape to simulate the background trend of the LOD. Maybe, as the reviewer thought, “as spectral properties of such series is specific and well distinct from that of sine curves”, however, if the longer period trend is characterized by various sine waves rather than the “saw-toothed” trend, then we still can nicely recover the target 6yr and 8.6yr signals (see the SI of this revised manuscript). As long as the frequencies of the two signals are different, we can separate these two signals and extract them respectively using the method of this work (i.e., NMWT+BEPME strategy), since NMWT method is just an approach to extract the target signal with the specific frequency (e.g., Wang et al, 2018; Duan and Huang 2019), which differs from the typical band-pass filtering methods.

Second, according to the suggestion provided by the reviewer, we will test the AR-1 stochastic series with significantly larger r.m.s. amplitude, with a decay time (240000 months) long compared with 6 yr.

Here, the AR-1 stochastic model can be written as (e.g., Gillet et al, 2017; Gillet et al, 2019):

$$d\xi + \xi \frac{dt}{\tau_u} = d\zeta(t) \quad \text{alternatively} \quad \frac{d\xi}{dt} + \frac{1}{\tau_u} \xi = \eta(t) \quad (4)$$

where, τ_u is called as the damped characteristic time, here it is set to be 240000 months, and

$\eta(t) = \frac{d\zeta(t)}{dt}$, which is the excitation series.

In mathematics, the formula (4) expresses a typical one-order differential equation, we can give its solution in the term of convolution form

$$\xi(t) = \eta(t) * e^{-\frac{t}{\tau_u}} = e^{-\frac{t}{\tau_u}} \int_0^t \eta(\tau) e^{\frac{\tau}{\tau_u}} d\tau \quad (5)$$

Formula (5) shows that the “output series” (i.e., $\xi(t)$) of AR-1 model also depends on the excitation term $\eta(t)$ - the input series. If $\eta(t)$ is an arbitrary stochastic “white-noise” series, we will get a stochastic oscillation AR-1 series $\xi(t)$. In fact, the formula (5) acts as a “low-pass” filtering process, where the function of $e^{-\frac{t}{\tau_u}}$ performs as a “low-pass filter”, whose filtering ability, however, is very poor, which may cause a frequency-spectral leakage to affect the high-frequency band.

Here, on the basis of the suggestion from the reviewer, we give a simulation test as following:

Firstly, we simulate a damped 6yr periodic oscillation (Figure 9a);

Secondly, using the above formula (5), we obtain an AR-1 stochastic series (i.e., the black curve in Figure 9b); furthermore, we obtain a larger AR-1 stochastic series through 5 times of the above AR-1 series (i.e., the green curve in Figure 9b) as the reviewer suggested;

Thirdly, adding the simulated damped 6yr signal to the above larger AR-1 stochastic series; the result (i.e., the composite series) is shown in Figure 9b (the red curve);

Finally, using the NMWT method to the above composite series; the result is shown in Figure 10a,b (i.e., the NMWT time-frequency spectrum).

Figure 10 shows that, in this simulation case, there is only one obvious spectral line in the NMWT spectrum. Of course, Figures 9-10 only shows one possible example of the various possible cases, furthermore, we give the following several cases to illustrate the relevant problems. Here, we use different input purely stochastic “white” series (i.e., $\eta(t)$) to obtain the various AR-1 stochastic series from the formula (5), and the relevant results are shown in Figure 11,

which presents some weak “periodic components” existing in the NMWT time-frequency spectrum besides the original 6yr periodic signal. The time-frequency domain results presented in Figure 11 mean that there might be more than one spectral lines in the NMWT time-frequency spectrum, which is attributed to the AR-1 continuously stochastic model, especially, when the magnitude of the AR-1 stochastic process is set to be large enough (e.g., $\times 6,7,8, \dots$).

Figure 9 This figure only show one possible case, since the different result (i.e., output series) can be obtained with the different “input” series- $E(t)$.

Figure 10 NMWT time-frequency spectrum of the above composite series

Figure 11 NMWT time-frequency spectrum of the above composite series

Although some of the above simulation tests appear to present the possible “multiple-lines” case (e.g., cases 3,4 in Figure 11), there are some questions worthy of further clarifying: 1) Whether the actual LOD variation necessarily satisfies the AR-1 model? 2) If the AR-1 model does apply, then whether the excitation $\eta(t)$ is necessarily the any purely stochastic continuous excitation series as the above simulation assumed? 3) whether such purely stochastic series is a sufficient condition to cause the corresponding LOD variations?

Of course, if all the answers to the above questions are positive, then a possibility that the target intradecadal variations (e.g., the normal mode signals) is likely polluted by this AR-1 stochastic process may exist, since in some of the simulation cases, some weak signatures may

appear on the 8~10yr scales. Here, we have also noticed the work of Gillet et al (2017) about the assumption that the stochastic AR-1 modelling forcing distributes within the bulk of the fluid outer core, nevertheless, in our opinion, why this stochastic excitation forcing will distribute within the bulk of the fluid outer core in physics? And why this excitation forcing respects the AR-1 purely stochastic model expressed by formula (4) ? If we only rely on AR-1 stochastic random excitation function model to explore the actual LOD variations, there will be additionally (at least) two problems required to be solved

Firstly The simulation results (i.e., the AR-1 stochastic series with a larger magnitude (e.g., $\times 5$ or greater) displayed in Figures 9-11) do not need to consider any Earth normal mode signals. As Gillet et al (2017) indicated, these results purely come from the stochastic excitation process without resorting to the other mechanisms, however, the actual $\eta(t)$ itself in formula (5) should be not the pure stochastic series, since $\eta(t)$ itself may own various eigen-frequencies (e.g., Buffett and Mound, 2005; Buffett et al, 2009).

Secondly The above tests only belong to the random trials, that is to say, if we do infinite simulation tests using an infinite number of significantly large AR-1 stochastic series, then we should be able to find an “output” result which looks like the actual LOD changes, meanwhile, we can give this corresponding “input” sequence (i.e., a purely stochastic series), but the question is that whether this purely stochastic “input” sequence is necessarily the real "input" excitation sequence of the actual Earth system? Which needs to further discussion.

In addition, as to the question that whether the intradecadal variations or the normal mode signals existing in LOD could be necessarily polluted by the AR-1 stochastic excitation? Which is beyond the scope of this work. But, even though the subdecadal normal mode signals existing in the LOD variations are really polluted by the above stochastic process, which still cannot negate the results detected by this work, since this work focuses on reporting that the intradecadal variations in LOD present two principle components (i.e., 6yr and 8.6yr) rather than the currently well accepted a 6yr component alone, and the phenomena that the 8.6yr component is increasing and its close relation with the geomagnetic jerk timings. Additionally, this work also attempts to combine the findings of this work with the current relevant published works (e.g., Mound and Buffett, 2003; Mound and Buffett, 2006; Gillet et al, 2010; Holme and de Viron, 2013; Gillet et al,

2015; Gillet et al, 2017; Chao 2017; Duan et al, 2018; Aubert and Finlay, 2019) to give a possibly reasonable explanation about the physical origins of these periodic components (especially the 8.6yr oscillation) detected by this work.

Of course, according to the above discussion and the previous works (Gillet et al, 2017; Gillet et al, 2019), we have further taken into account the above simulation results and discussed them in the revised manuscript, please see the revised version, page 11, lines 261th~265th. In summary, in our opinion, the reviewer provides a good and constructive suggestion for us, and we would like to further consider the issues of the stochastic excitation model (e.g., Gillet et al, 2017) in our future work. Thank the reviewer very much for the good suggestion once again.

This way, the author could furthermore test if they recover the decay rate associated with the « true » damped oscillator series. Finally the same test could be performed by considering two damped oscillators instead of one.

Response: Thank the reviewer very much for the good suggestions. The result recovered by our method (i.e., NMWT+BEPME) used in this manuscript actually depends on the time-varying characteristic of the original oscillation signal. That is to say, if a 6yr decaying oscillation signal (here, the decaying information is unknown in advance) exists in the original observed series (e.g., the observed LOD data), then the result recovered by NMWT+BEPME method will be a decreasing signal as well. Nevertheless, a periodic decreasing oscillation signal may include two cases: 1) a free damping oscillation (i.e., the theoretical oscillation curve, see Figure5(b)), 2) the consequence of a continuous stochastic excitation (see Figure 5(c)). If it is just the former case, then we can nicely recover the real free damped oscillation (in this case, the observed quality factor Q value equals to the theoretical value, where the Q value estimated from the damping oscillation result recovered by the NMWT+BEPME is called as the observed Q value), we have demonstrated this point in our previous works (e.g., Duan et al,2017; Duan and Huang 2019, and also in this manuscript). However, if it is the latter case, then the “output” series via the continuous stochastic excitation may be various with the different “input” series (e.g., Figure 7), that is to say, if the “output” result (or the original observed signal) is decaying, then the result recovered by our method will be decreasing as well, nevertheless, at this time, the wavelet result may not reflect the free damping oscillation, which is similar to the Chandler wobble (CW), and in

this case, the observed Q value may be no longer its theoretical Q value, therefore, in this case, the decaying oscillation recovered by the NMWT method does not necessarily reflect the actual free damped oscillation.

Despite this, we would like to mention the advantage of NMWT method once again: even though we do not know the prior information (e.g., the decaying or increasing) of the target signals in advance, we still can recognize this decaying (or increasing) signal and recover it perfectly in the time domain (e.g., Duan and Huang, 2019) through using the NMWT+BEPME method. As Duan et al (2017) indicated, if the observed 6yr oscillation in LOD is consequence of the significantly continuous stochastic excitation, then the current $Q \sim 51$ estimated by Duan et al (2017) may be not the theoretical value of the 6yr free normal mode signal. Therefore, we call this value Q (~ 51) as the observed quality factor in this work (of course, also in Duan et al, 2018), which is similar to the Chandler wobble in the polar motion. Of course, as Duan et al (2017) discussed, we agree with the idea that the observed 6yr oscillation should be the result of excitation, however, the question is that, up to now, we have not found the strong evidence to demonstrate that the observed 6yr oscillation in LOD must be subject to the significantly continual stochastic excitation during 1962~2019, since, up to now, we have not found the reliable stochastic excitation series (or excitation events). Conversely, the 6yr oscillation time-domain result recovered by the NMWT method can be nicely characterized by a typical exponential decaying function (see Duan et al, 2018), furthermore, the strength of electromagnetic coupling (i.e., $\sim 38\text{Nm}^{-3}\text{s}$) at the core-mantle boundary inferred by this observed quality factor value (i.e., $Q \sim 51$) is also acceptable (e.g., Gillet et al, 2010), so we gave a possible explanation that the current 6yr signal decaying during the past several decades (i.e., 1962~2012) detected by Duan et al (2015) is due to the electromagnetic coupling dissipation at the core-mantle boundary (Duan et al, 2017; Duan et al, 2018).

Additionally, this work does not mainly focus on the issues about the 6yr oscillation in LOD, though this 6yr oscillation issues are still worthy of further study later. According to the above discussion, we have further added the relevant discussion in the revised manuscript, please see the revised version, i.e., pages 4~5, lines 100th~110th.

As to the two damped oscillation cases: If these two oscillation signals are free decaying oscillations, then NMWT+BEPME can well recover them respectively in the time domain (e.g.,

Duan and Huang,2019). Furthermore, the above simulation tests have shown that NMWT method owns a high frequency-resolution, which can clearly distinguish the 6yr, 7.2yr and 8.6yr components. If these two signals (i.e., 6 yr and 8.6 yr) are the damped stochastic oscillations which are presented in the Figure 12, we can recover these decaying information on the original signals using NMWT+BEPME method and further use these recovered attenuated results to further infer the relevant excitation information. In summary, combining the NMWT method with the BEPME strategy is indeed an effective harmonic analysis approach to obtain a good recovered result (including the decaying or increasing information, the correct phase information) of the target monochromatic signal from the original data series.

In addition, according to the simulation tests (e.g., the “output 9” series presented in Figure 12), we do not exclude a possibility that the relatively weaker signal during the 6yr period and the 8.6yr period (e.g., the “quasi-7.2yr signal”) presented in the LOD spectrum may be the consequence of the stochastic excitation of two normal mode signals (e.g., the 6yr and 8.6yr oscillations). According to the above discussion, we have further revised the manuscript, please see the revised version, i.e., page 3, lines 65th~70th; and the pages 19~23, lines 364th~438th in “supplementary information” of this revised manuscript.

Figure 12 The left columns show the various “output” series (6yr damped stochastic oscillation+8.6yr damped stochastic oscillation), the middle and the right columns show the NMWT time-frequency spectrum of the “output” series, where the horizontal ordinates represent the time/month, while the vertical coordinates express the period/year.

I also ask these questions since SSA on LOD series cleaned from atmospheric winds and solid tides contributions clearly show a 6 yr line, but not the one at 8.5 yr.

Response: Thank the reviewer very much for the good question. In our previous works (e.g., Duan et al, 2015; Duan et al, 2017), we did not use the singular spectrum analysis (SSA) method to analyze the LOD data. Here, considering the reviewer raised this question, we use a typical simulation analysis to illustrate the relevant issues of using SSA method. Through the following simulation test, SSA method is not an ideal approach to distinguish the intradecadal components existing in the LOD variations.

As the previous works (e.g., Chen et al, 2013; Wang et al, 2018) indicated, the frequency-resolution of SSA method is related to the window length parameter (L), choosing an appropriate L value is important for SSA method to analyze the actual series. As Chen et al (2013)

indicated, L should neither be too large nor too small. If L is too small, the coarse resolution may cause several neighboring peaks in the spectrum of signal to appear as one, on the other hand, large L values will split the peak into several components with neighboring frequencies. Here, we will use a simulation test and an actual LOD data to answer the above question.

We simulate a composite signal of the superposition of three harmonic components:

$$Y(t)=y_1(t)+y_2(t)+y_3(t)+\text{noise}(t)$$

where, the data length is 686 months, which is the same length as that of the currently actual observed LOD data; here, y_1 , y_2 and y_3 are respective the 6yr (decaying), 7.2yr (steady) and 8.6yr (increasing) signals, and their expressions are following:

$$\begin{aligned} y_1 &= 0.12 \exp(-0.001t) \sin(2\pi f_1 t); & f_1 &= 0.0138 \text{cpm} & \% \text{ a decaying 6yr signal} \\ y_2 &= 0.05 \sin(2\pi f_2 t); & f_2 &= 0.01157 \text{cpm} & \% \text{ a steady 7.2yr signal} \\ y_3 &= 0.05 \exp(0.0009t) \sin(2\pi f_3 t); & f_3 &= 0.00969 \text{cpm} & \% \text{ an increasing 8.6yr signal} \end{aligned}$$

The above $Y(t)$ series and the various corresponding SSA results are presented in Figure 13a. Furthermore, we show the SSA results in frequency domain, see Figure 13b. Here, we need to discuss how to choose the appropriate positive integral parameters of SSA method, i.e., L and I , where I reflects the so-called decomposition level, when I is set to be larger, the higher-frequency signal will be obtained, otherwise the opposite:

Firstly, if L is invariable (here $L=600$), then the SSA results (as I equals to 1 alternative 2) can well characterize the above simulated composite signal- $Y(t)$, see the Figure 13a; when $I \geq 3$, the SSA results are tend to be non-ideal to characterize the target composite signal (Figure 13a). Consequently, we only need to use $I=1$ (or 2) to analyze the LOD intradecadal variations based on the SSA method. Meanwhile, Figure 13b indicates that the frequency-domain results of SSA method (as parameter $I=1$ or 2) present a wide energy-spectrum within the 5~10yr band rather than a 6yr sharp peak. Consequently, these SSA results cannot well distinguish the target signals (i.e., the 6yr, 7.2yr and 8.6yr) existing in the original composite signal.

Secondly, if I remains to be invariable (here $I=2$), when the parameter L is set to be relatively large (e.g., $L=600,550$), SSA results show a wide energy-spectrum (Figure 14a,b); when L is relatively small (e.g., 500, 450 and smaller), then SSA results will trend to be a single 6-yr peak (Figure 14b), this corresponding time-domain result (Figure 14a) from SSA method, however, is inconsistent with the original simulated 6yr decaying oscillation. In addition, this single 6yr peak

does not mean that only a 6yr signal exists in the original composite series, since the original simulated series contains three frequency components.

Consequently, the above simulation analysis shows that although SSA method may well recover the original superposition signal $Y(t)$ when the parameter L and I are adopted to be the appropriate values (Figure 13a, Figure 14a, e.g., $L=600$, $I=2$), but it cannot effectively distinguish the above target 6yr, 7.2yr and 8.6yr components due to the frequency-resolution issue of the SSA method. The time domain result from SSA method ($L=600$, $I=2$) shows a modulation phenomenon, which actually attributes to the interaction of the various components within the 5~10yr band (Figure 13b, Figure 14b).

Figure 13. Comparison of the SSA result and the original simulated series in both time and frequency domains, here $L=600$.

Figure 14. Comparison of the SSA result and the original simulated series in both time and frequency domains, here $I=2$.

Furthermore, applying SSA method to the actual LOD intradecadal variations. According to the above simulation, we can use the relevant parameters (i.e., $L=600, 550, 500, 450$, while $I=2$) to

analyze the actual LOD intradecadal variations based on the SSA method, and the results (in both time and frequency domains) of which are shown in the Figure 15.

Comparing the Figure 14 with the Figure 15, a wide energy spectrum range of LOD variations recovered by the SSA method is shown in Figure 15b (when L value is relatively larger, e.g., $L=600$ or 550), which shows that the actual LOD intradecadal variations should not contain a 6yr component alone. When L is adopted to be relatively smaller value (e.g., $L=450$), the spectrum of LOD variations recovered by the SSA method is shown to be a 6yr peak, which does not mean that the actual LOD variations only contains a 6yr oscillation alone within the 5~10yr band.

Figure 15. Comparison of the SSA result and the original LOD intradecadal variations in both time and frequency domains.

In summary, through this simulation analysis, this case that “SSA on LOD series cleaned from atmospheric winds and solid tides contributions clearly show a 6 yr line, but not the one at 8.5yr” suggested by the reviewer indicates that SSA method is not an ideal approach to quantitatively distinguish and accurately isolate the target harmonic components existing in the LOD intradecadal variations. However, NMWT method can clearly distinguish the target signal components, for example, see Figure 4 .

According to the good question and the above discussion, we have further revised the manuscript, please see the revised version, i.e., page 3, lines 79th~80th; page 14, lines 362th ~366th; in the supplementary information, pages 5~8, lines 99th~166th.

Finally, why is this new line not apparent in previous studies implying the authors [8,13] ? is it only because the spectra were cut in Figures at periods shorter than 8 yrs ?

Response: Thank the reviewer for the good questions. Because our previous works (e.g., Duan et al, 2015; Duan et al, 2017) mainly focused on the 6yr oscillation, and at that time, we have not concerned about the lower-frequency harmonic signals on the intradecadal scales too much, so that we only set the period $T < 7\text{yr}$ in our program codes of the NMWT method, thus the 8.6yr signal have not shown in our previous work (see the Figure 3 and Figure 4 in the work of Duan et al, 2015). Yes, it is only because the NMWT spectrum were cut in Figures at periods shorter than 8 yrs in our previous studies.

2) jerks catalogues :

The authors refer to jerks dated from [6, 23, 24]. There exists two systematic studies of such events that the authors shall consider: Brown et al (2013) and Soloviev et al (2017). In the former, more frequent jerks are found as data become more accurate, almost filling the entire span at recent epochs. In the second, SA pulses (in between jerks) are found at preferred epochs, with consequences on the dates of jerks (some being less v-shaped than others). Jerks dates from these two studies should be considered when building and discussing Fig. 4.

Response: Thank the reviewer very much for the good suggestions. According to these suggestions, we have further discussed the previous works of Brown et al (2013) and Soloviev et al (2017) in the revised manuscript. For example, Brown et al (2013) used the monthly mean geomagnetic data to discuss the geomagnetic jerk occurrence and their uncertainties, and they found the jerk abound feature. Although the references (Holme and de Viron, 2013; Chulliat and Maus, 2014; Torta et al, 2015) did not present the existence of 1995 jerk, Malkin (2013) indicated that a potential jerk event occurred in ~1995 (i.e., the “1995 jerk” in Figure 4). Meanwhile, Brown et al(2013) also showed the jerk event happened during 1995~1998, though these jerk span was shown almost filling the entire span at recent epochs. In addition, Soloviev et al (2017) indicated that SA pulses (in between jerks) are found at preferred epochs, which shows that the jerk occurrence may own a certain periodic feature, while other works (e.g., Chulliat and Maus, 2014; Torta et al, 2014) also found the jerk occurrence has periodic characteristics, and suggested that

mechanism response for the jerks may be related to a certain periodic oscillation (e.g., Torta et al, 2014), the Figure 4 in this work supports an observed evidence of this viewpoint.

According to the above discussion, we have further revised the manuscript, please see the revised version, page 8, lines 182th~186th, and lines 190th~194th.

Furthermore, there is currently a strong debate on the existence of jerks localized in time, which should be recalled in this study. The localization of jerks in time could be the effect of considering filtered series (because we cannot separate well internal from external sources). It has some implications on the spectra of SV series. Indeed, SA pulses seem to occur at preferred epochs (see Soloviev et al), but SV series do not show clear isolated spectral lines at subdecadal periods – see for instance to Gillet (2019) and to Lesur et al (2017). The sentence « the above phenomenon... » on p. 6 should be modified accordingly.

Response: Thank the reviewer very much for the good suggestions. According to these suggestions, we further recalled the relevant debate on the existence of jerks localized in time in the revised manuscript. The sentence « the above phenomenon... » in the original manuscript has been modified in the revised version. Because we cannot separate well internal from external sources, it has some implications on the spectra of SV series, thus there is currently a strong debate on the existence of jerks localized in time. Nevertheless, Pinheiro et al (2011) specially studied the following four well-known jerks (i.e., 1969, 1978, 1991, 999), while these jerk timings are all well correspondence with the extremes of the 8.6 yr oscillation recovered by this work. Furthermore, Holme and de Viron (2013) also listed the catalogue of the jerk timings during 1962-2007, these timings are thought to be the best determination (see Figure 2 in the work of Holme and de Viron, 2013).

According to the above suggestions and discussion, we have revised the manuscript, please see, page 7, lines 167th~177th; page 8, lines 189th~194th.

3) geophysical consequences :

- The discussion at the end of p. 6 should be strongly rephrased. I do not see why « Fig. 4 also reveals that the increasing... these jerks ». This whole paragraph seems highly speculative.

Response: Thank the reviewer very much for the good suggestions. According to the suggestions, we have further revised the discussion at the end of p. 6 in the original manuscript. In addition, we add some discussion about the issue that how the possible mechanism responsible for the jerks to cause the 8.6yr oscillation amplitude increasing. We have revised the whole paragraph, please see the revised manuscript, for example, page 8, lines 189th~194th.

- The authors claim (p. 7) that the 6 yr LOD changes is « most likely » excited by an inner core swing. This sentence is not correct. This is one possibility among others. The inner core is only responsible for a tiny fraction of the angular momentum budget, and the LOD series alone cannot be considered as a proof of inner core oscillations.

Response: Thank the reviewer for the good comments. We agree with the idea proposed by the reviewer that the inner core swing is one possibility among other mechanisms responsible for the observed 6-year oscillation in LOD. Here, we would like to further discuss this possibility. Although the inertia moment ($C_i \sim 5.87 \times 10^{34} \text{kgm}^2$) of the inner core (IC) is indeed much smaller than that ($C_m \sim 7.12 \times 10^{37} \text{kgm}^2$) of the mantle, which means that the angular momentum budget from the IC should be small, however, the observed amplitude of the 6yr oscillation in LOD is not great (only $\sim 0.12 \text{ms}$), in other words, this small angular momentum budget from the IC is large enough to explain this observed amplitude of the 6yr oscillation.

Here, we only consider the gravitational coupling interaction between the mantle and the IC without involving the other coupling effects (e.g., electromagnetic coupling, viscous coupling). If assuming that the observed 6yr oscillation is fully attributed to the MICG mode, the magnitude of the 6yr oscillation in LOD is related to the axial rotation angular velocity of the IC departing from the gravitational equilibrium position (e.g., Buffett, 1996; Mound and Buffett, 2003, 2006; Chao 2017; Duan et al, 2018). Under the condition of the angular momentum conservation between the IC and the mantle, we can build the relationship between the amplitude of the 6yr oscillation and the axial rotation angular velocity of the IC.

According to the angular momentum conservation law, at any time t , the IC axial rotation angular velocity $u_i(t)$ and the angular velocity of the $u_m(t)$ satisfies the following relationship

$$u_m(t) = -\frac{C_i}{C_m} u_i(t) \quad (1)$$

According to the relationship between $u_m(t)$ and the LOD variations (i.e., ΔLOD)

$$u_m(t) = -\frac{2\pi}{(LOD_0)^2} \Delta LOD(t)$$

so

$$u_i(t) = \frac{2\pi}{(LOD_0)^2} \frac{C_m}{C_i} \Delta LOD(t) \quad (2)$$

When $|\Delta LOD(t)|=0.12\text{ms}$, we can obtain the amplitude of $u_i(t)$ is $0.22^\circ/\text{yr}$. Importantly, this IC rotation rate ($\sim 0.22^\circ/\text{yr}$) required by the 6yr oscillation is compatible with the IC rotation rate inferred by the seismology, for example, seismic normal mode estimated this rate is $\pm 0.2^\circ/\text{yr}$ (Lask and Master, 1999), while the earthquake doublets indicates that this rate is $0.25\sim 0.48^\circ/\text{yr}$ (Tkalcic et al, 2013).

According to the good comment and the above discussion, in the revised manuscript, we have further discussed the issue of angular momentum budget under the MICG interaction, please see the revised manuscript, page 9, lines 223th-229th; pages 15-16, lines, 386th~404th.

On the other hand, angular momentum associated with TW alone explains subdecadal LOD changes, and there is yet no definitive scenario for their triggering. It may be associated with a swing of the inner core, but alternatively Lorentz torques on the the inner core, or within the bulk, could equally well generate waves travelling from the inner core (see Teed et al 2014 ; Gillet al al, 2017).

Response: We agree with this comment and add it into the revised manuscript. Please see the revised version, i.e., page 9, lines 232th~234th. That is, “TW may explain the subscadedal LOD changes through transferring the angular momentum from FOC to the mantle. Nevertheless, there is yet no definitive scenario for their triggering. It may be associated with a swing of the inner core, but alternatively Lorentz torques on the the inner core, or within the bulk of the fluid out core, could equally well generate waves travelling from the inner core (see Teed et al 2014 ; Gillet al al, 2017).”

- The authors claim (p. 8) that a period of 8.6 yrs for torsional waves (TW), instead of 6 yrs, should lead to revise down the strength of the magnetic field deep in the core. This is without considering that the period is not immediately related to the field intensity. Actually this latter is constrained by the slowness (integral, along the cylindrical radius, of the inverse of the Alven speed, see Gillet et al, 2017). This latter defines the waveform, and it is the waveform that leads to the estimate of the field intensity in the core by [1]. As a consequence, one cannot follow the argument on a lower field intensity.

Response: Thank the reviewer for the good comments and suggestions. We are very sorry that, in the original manuscript, we misunderstood the concept of the magnetic field intensity related to the period of the torsional waves within the fluid outer core. According to these nice suggestions, we have further corrected our manuscript and discussed the cylindrical radial component of the magnetic field ($\tilde{B}_s(s)$) within the fluid outer core based on the works (e.g., Gillet et al,2010; Gillet et al, 2017).

The cylindrical radial component of the magnetic field ($\tilde{B}_s(s)$) inside the FOC from the eigenperiod formula (Gillet et al, 2010; Gillet et al, 2017; Finlay et al, 2010) of the FOC torsional oscillation normal mode, instead the 8.6-year period should be used to infer $\tilde{B}_s(s)$ with the following formula

$$\tilde{B}_s(s) \approx \frac{r_f}{\tau} \sqrt{\rho_0 \mu_0} \quad (1)$$

where, $\tilde{B}_s(s) = \sqrt{\frac{1}{4\pi h} \int_{-h}^{+h} \int_0^{2\pi} B_s^2(s, \phi, z) d\phi dz}$; here $h(s) = \sqrt{r_f^2 - s^2}$ is the half-height of a

fluid cylinder, and the cylindrical (s, ϕ, z) coordinates is adopted, i.e., s is the radius of the cylinder, ϕ expresses the longitude, z direction is aligned with the Earth rotation vector $\vec{\Omega}$.

We have further revised the manuscript, please see the revised version, i.e., page 10, lines, 237th~244th.

- Next the authors mention (still on p. 8) that waves in a stratified layer have been « detected » ; this is not correct. What Chulliat et al (2015) have shown is that one may possibly interpret subdecadal SV changes as the signature of such waves. This paragraph is speculative, especially since in such a case, associated surface core flows would not explain directly LOD changes anymore – see also the recent work by Gastine et al (2019) that does not favor the presence of a stratified layer at the top of the outer core.

Response: Thank the reviewer for the good suggestions. According to this suggestion, we have modified the original manuscript and added the argument of the work (Gastine et al,2019) after the discussion of this issue. Please see the revised manuscript, pages 10, lines, 253th~257th.

- Alternatively, the authors should discuss further the work by Aubert & Finlay (2019), in particular the link these authors see between jerks and the acceleration of zonal motions (and thus possible links to LOD variations, as in the study by Holme & de Viron, 2013).

Response: Thank the reviewer for the good suggestions. We have further discussed the work of Aubert & Finlay (2019) in the revised manuscript. It is quite plausible for the geomagnetic jerk to spur the amplitude increasing of the LOD variations (Holme and de Viron, 2013; Aubert and Finlay 2019). According to a current numerical simulation analysis (Aubert and Finlay 2019), the geomagnetic jerks can be induced by the arrivals of localized Alfvén wave packets. As these waves reach the surface of the fluid core, they focus their energy towards the equatorial plane and along the strong magnetic flux lines, making the sharp interannual core flow change. That is, these jerks can induce the acceleration of the azimuthal flow motions (Aubert and Finlay 2019), which associates with the angular momentum exchanges between core and mantle (e.g., Holme and de Viron, 2013), and thus to excite the LOD variations, while the result of this work further shows that these excitation events related to the geomagnetic jerks always occurred near the timings of the 8.6yr oscillation extremes (see Fig.4 in the manuscript) during the past several decades, causing the amplitude of the 8.6yr signal to continuously increase.

According to the suggestions, we have further revised the manuscript, please see the page 11, lines 272th~281th.

- Finally, the author mention the « obvious » periodicity in jerk occurrence (see p. 9). This statement should be softened (see above point 2). They also speculate on the possibility to predict jerks. This idea should be further developed in the light of the results by Aubert & Finlay (2019) ; also a discussion of the analysis of dynamo simulations by Gillet et al (2019) would be welcome. These authors find that a large fraction of rapid SV changes are due to unmodelled SV associated with subgrid-scale induction... how predictable are these, if not correlated with large length-scale dynamics ?

Response: Thank the reviewer for the good suggestions. According to these suggestions and some discussion of the results by Aubert and Finlay (2019) as well as the analysis of dynamo simulation by Gillet et al (2019) are also made and used to interpret the relationship between the 8.6yr oscillation and the jerks occurrence, we further soft the relevant statement. We do not excluded the possibility that the jerk occurrence owns a certain randomness. Given some works (e.g., Chulliat et al 2014; Torta et al, 2015) showed that the jerk occurrence owns some periodic feature due to some periodic physical mechanism through analyzing the current geomagnetic field data, while Aubert & Finlay (2019) shows that jerks can induce the acceleration of the equatorial interannual zonal flow motions through the geodynamo numerical simulation analysis, the signatures of which appears to be detected by Gillet et al(2019) at the low latitude region within $\pm 10^\circ$. Because of these acceleration flow can transfer angular momentum from fluid core to the mantle, while during the past several decades, this excitation always occurred near the extremes of the 8.6yr oscillation, causing its amplitude increasing. As Aubert and Finlay (2019) indicated, jerks represent a major obstacle to predict the geomagnetic field behaviors for yeas to decade ahead, we hope that this work can provide a new entry to predict the jerks through the relationship between the jerk events and the 8.6yr oscillation in LOD.

According to the suggestions and the above discussion, we have further revised the manuscript, please see the pages 11-12, lines 288th~293th.

Minor points :

- p. 2 : the last sentence seems rather speculative at this stage. It should be removed. There is no definitive evidence of a triggerring of TW by oscillations of the inner core.

Response: According to this suggestion, we have remove the last sentence in the revised manuscript. please see the revised manuscript, page 3, lines 83th~85th.

- p. 6 : why refer to [22] here ? There exists more accurate references.

Response: Thank the reviewer for the good suggestion. We have revised the manuscript, please see the revised version.

- p 8 : why refer to [29] for the excitation of torsional waves on the inner core ? There are better references (e.g. Mound & Buffett, 2006).

Response: Thank the reviewer for the good suggestion. We have revised the manuscript, please see the revised version.

- method section : isn't it an ambiguity when extending with the symmetric series in a case where there are 2 sine functions ? I guess that if one is larger than the other it may work, but what happens

Response: Thank the reviewer for the good question. In the time domain, it appears to be an ambiguity, however, the “BEPME” strategy adopted in this work does not extend the original temporal series directly in the time domain. Because the NMWT belongs to the time-frequency analysis approach (e.g., Liu et al, 2007; Duan et al, 2015; Duan and Huang, 2019), while in such case that there are 2 sine functions, these two sine functions can be separated clearly in the NMWT time-frequency domain (for example, see Figure 4). In fact, we are to extract them respectively using the NMWT+BEPME strategy, and the recovered results are not influenced by the relative amplitudes of the two sine signals. Here, we give a typical simulation analysis to demonstrate this point. We use the following composite signal $Y(t)$ comprised by two sine functions (i.e., 6yr and 8.5yr) with the same amplitudes

$$Y(t)=y_1(t)+y_2(t)+noise(t)$$

where,

$$y_1(t)=1.5\sin(2\pi f_1 t), f_1=0.0138\text{cpm}, \quad \% \text{ the 6yr period sine signal}$$

$$y_2(t)=1.5\cos(2\pi f_2 t+\pi/3), f_2=0.0098\text{cpm}, \quad \% \text{ the 8.5yr period sine signal}$$

here, the $y_1(t)$, $y_2(t)$ and $Y(t)$ are displayed in Figure 16.

Firstly, we directly use NMWT to the above composite signal $Y(t)$, and we obtain the NMWT time-frequency spectrum shown in the Figure 17; from the spectrum, we can clearly find the two periodic components existing in $Y(t)$ and the corresponding two“ridge-lines”;

Secondly, we directly extract the signals locating at the “ridge lines”, and the results are shown in Figure 18; because of the edge effect, at this time, the results are not accurate in amplitudes (also can be seen in the Method of this manuscript); despite this, the phase of the recovered signals are accurate enough, which has been shown in Figure 18.

Thirdly, using the BEPME strategy to extend the series (how to process this, please see the manuscript), then using NMWT to analyze this extension time series, and the results are shown in Figure 19, which shows that this method (NMWT+BEPME) can recover the target signals perfectly.

This simulation analysis shows that even though the original two signals have the same amplitude, we still can recover them perfectly. In addition, why we need to extend the series at the extreme points of the target harmonic signals? The purpose of which is to make sure (as much as possible) that the target signal curves at the extension points are continuous and smooth, otherwise, the recovered results may be non-ideal outcomes.

Figure 16 a shows the simulated 6yr period signal; b indicates the 8.5yr signal; c displays the composite signal- $Y(t)$.

Figure 17 The NMWT time-frequency spectrum of the composite signal- $Y(t)$; the white lines are just the “ridge lines” (e.g., Duan et al, 2015)

Figure 18 Comparison of the results directly from the NMWT (without the “BEPME” processing) with the simulated target signals, which indicates that although edge effect can significantly influence the amplitudes of the target signals, the phase information are unbiased, which can make us successfully search for the moments (i.e., the black vertical dashed lines) of the extreme points of the target signals at the two boundaries, and further removing the data outside the moments of the extreme points.

Figure 19 Comparison of the results from the NMWT+BEPME adopted in this work and the simulated original signals, which further shows that the method (i.e., NMWT+BEPME) is a good approach to recover the target harmonic signals.

References

- Aubert, J., Finlay, C.C. Geomagnetic jerks and rapid hydromagnetic waves focusing at Earth’s core surface. *Nature Geoscience*, 12, 393-398 (2019).
- Brown, W.J., Mound, J.E., Livermore, P.W. Jerks abound: An analysis of geomagnetic observatory data from 1957 to 2008. *Phys. Earth Planet Inter*, 233, 62-76 (2013).
- Buffett, B.A., Mound, J.E., 2005. A Green’s function for the excitation of torsional oscillations in the Earth’s core. *J. Geophys. Res.* 110,B08104,doi:10.1029/2004JB003495.
- Buffett, B.A., Mound, J., Jackson, A., 2009. Inversion of torsional oscillations for the structure and dynamics of Earth's core. *Geophys. J. Int.* 177,878-890. doi: 10.1111/j.1365-246X.2009.04129.x.

- Chao, B F. Dynamics of axial torsional libration under the mantle-inner core gravitational interaction, *J. Geophys. Res. Solid Earth*, 122, 560-571 (2017).
- Chen, Q., Van Dam, T., Sneeuw, N., Collilieux, X., Weigelt, M., Rebischung, P. Singular spectrum analysis for modeling seasonal signals from GPS time series. *J. Geodyn.* 72, 25-35 (2013).
- Chulliat, A., Alken, P., Maus, S. Fast equatorial waves propagating at the top of the Earth's core. *Geophys. Res. Lett.*, 42, 3321-3329 (2015).
- Chulliat, A., Maus, S. Geomagnetic secular acceleration, jerks, and localized standing waves at the core surface from 2000 to 2010. *J. Geophys. Res: Solid Earth*, 119, 1531-1543 (2014).
- Duan, P.S., Liu, G.Y., Liu, L.T., Hu, X.G., Hao, X.G., Huang, Y., Zhang, Z.M., Wang, B.B. Recovery of the 6 year signal in length of day and its long term decreasing trend. *Earth Planets Space.* 67:161, doi:10.1186/s 40623-015-0328-6 (2015).
- Duan, P.S., Liu, G.Y., Hu, X.G., Sun, Y.F., Li, H.L. Possible damping model of the 6 year oscillation signal in length of day. *Phys. Earth Planet. Inter.*, 265. 35-42 (2017).
- Duan P S, Liu G Y, Hu X G, Zhao J, Huang C L. Mechanism of the interannual oscillation in length of day and its constraints on the electromagnetic coupling at the core-mantle boundary. *Earth Planet. Sci. Lett.* 482, 245-252 (2018).
- Duan, P. S and Huang, C. L. Application of normal Morlet wavelet transform method to the damped harmonic analysis: On the isolation of the seismic normal modes (${}_0S_0$ and ${}_0S_5$) in time domain. *Phys. Earth Planet. Inter.*, 288, 26-36 (2019).
- Finlay, C.C., Dumberry, M., Chulliat, A., Pais, M.A. Short timescale core dynamics: theory and observations. *Space Sci Rev.* 155:177-218 (2010).
- Gillet, N., Jault, D., Canet, E., Fournier, A. Fast torsional waves and strong magnetic field within the Earth's core. *Nature.* 456, 74-77 (2010).
- Gillet, N., Jault, D., Canet, E. Excitation of travelling torsional normal modes in an Earth's core model. *Geophys. J. Int.* 210, 1503-1516 (2017).
- Gillet, N., Huder, L., Aubert, J. A reduced stochastic model of core surface dynamics based on geodynamo simulations. *Geophys. J. Int.* 219(1), 522-539 (2019)
- Gillet, N. Spatial and temporal changes of the geomagnetic field: insights from forward and inverse core field models. *Geomagnetism, Aeronomy and Space Weather: a Journey from the Earth's Core to the Sun* (2019).

- Holme, R., de Viron, O. Characterization and implications of intradecadal variations in length of day. *Nature*. 499, 202-205 (2013).
- Liu, L.T., Hsu, H.T., Grafarend, E.W. Normal Morlet wavelet transform and its application to the Earth's polar motion. *J. Geophys. Res.* 112, B08401, doi:10.1029/2006JB004895 (2007).
- Laske, G., Masters, G. Limits on differential rotation of the inner core from an analysis of the Earth's free oscillations, *Nature*, 402, 66-69 (1999).
- Mound, J.E., Buffett, B.A. Interannual oscillations in length of day: Implications for the structure of the mantle and core. *J. Geophys. Res.* 108 (B7), 2334, doi:10.1029/2002JB002054 (2003).
- Mound, J.E., Buffett, B. A. Detection of a gravitational oscillation in length-of-day. *Earth Planet. Sci. Lett.* 43, 383-389 (2006).
- Pinheiro, K.J., Jackson, A., Finlay, C.C. Measurements and uncertainties of the occurrence time of the 1969, 1978, 1991, and 1999 geomagnetic jerks. *Geochem. Geophys. Geosyst.*, 12, Q10015, doi:10.1029/2011GC003706 (2011).
- Soloviev, A., Chulliat, A., Bogoutdinov, S. Detection of secular acceleration pulses from magnetic observatory data. *Phys. Earth Planet. Inter.*, 270:128-142 (2017).
- Teed, R.J., Jones, C.A., Tobias, S.M. The dynamics and excitation of torsional waves in geodynamo simulation. *Geophys. J. Int.* 196, 724-735 (2014).
- Tkalcic, H., Young, M., Bodin, T., Ngo, S., Sambridge, M. The shuffling rotation of the Earth's inner core revealed by earthquake doublets. *Nature Geoscience*, 6, doi:10.1038/NGEO1813 (2013).
- Torta, J.M., F. J. Pavón-Carrasco, S. Marsal, and C. C. Finlay. Evidence for a new geomagnetic jerk in 2014, *Geophys. Res. Lett.*, 42, 7933–7940, doi:10.1002/2015GL065501 (2015).
- Wang, G.C., Liu, L.T., Xu, A.G., Pan, F., Cai, Z.W., Xiao, S.H., Tu, Y., Li, Z.H. On the capabilities of the inaction method for extracting the periodic components from GPS clock data. *GPS. Solution*, 22:92 (2018).

Reviewer #3 (Remarks to the Author):

Reviewer: Richard Holme

Note: this review is phrased in a rather personal way. This is not because I am upset that the authors are trying to supersede my work – more power to them! – but because what they are doing is antithetical to my approach – I seek simplicity, without relying on a *dux exmachina* from complicated maths – and I would argue that my methods achieve a better fit to data. Fundamentally, which is correct may not be answerable, although I suggest a test below.

This paper examines the variation in LOD. It is a development of an earlier paper (by me!) which argued that intradecadal variation in LOD could be modeled almost entirely by a 5.8 year period signal of constant amplitude. Such a model was not a perfect fit – there were clearly jumps in the LOD which can not be fit by a simple harmonic signal – but nonetheless, it explained most of the variance. One additional point in that paper is relevant to this work – the longer period decadal variation was removed by a smooth curve fit, and this fit was repeated iteratively after subtracting the harmonic variation – the variation requires detrending with the long period signal, while the long period signal required detrending the oscillation before it is fit.

This paper instead uses various rather more developed techniques (Morlet wavelets, BEPME (which I will not write in long form!)) to attempt to separate the longer period signal, then shows a good fit to the residual with wavelet methods, and then fits the wavelet signals with two different harmonic signals – one of about 6 year period and 8.6 year signal, the former decaying, the latter growing. The authors argue that this model better fits the data.

First I must say that this problem is strongly non-unique – both the earlier formalism and the work in this paper are possible fits to the data, and on some level it is not possible to judge which of the two is “better”.

Where the two methods differ is the differing reliance on complex methods. This paper relies on its methodology, and in particular a spectral separation – the background signal is obtained by a low pass filter, defining it in terms of its period. This would work if the signal was longer (so more than a few periods) or genuinely appeared stationary, but it does not, either at long periods or short periods. The remaining shorter periods are then fit very well by the wavelet transform – but this is inevitable, as the wavelets have in effect almost a continuum of free parameters. The final

argument for the growing and decaying exponentials, leading to a 6 parameter fit – two amplitudes two phases, two decay rates) does not fit the data particularly closely (see figure 3, and compare with figure S3 from my earlier paper for a fit with only a 2 parameter system).

Response: Thank professor Holme very much for providing the careful and constructive comments and suggestions on our manuscript. To our understanding of the above comments, the main concern is comparing the method (i.e., normal Morlet wavelet transformation- NMWT) used in this work with the iterative fit method (Holme and de Viron, 2013) with simple philosophy. How to confirm the efficiency and reliability of the proposed methods (i.e., the NMWT) in analysis of the current LOD data. In order to answer this main concern, we made some typical numerical simulations with various cases in which consisting of varying trend, various harmonic oscillations, significant noises, and the same length of the time series as the actual LOD data.

Firstly, we acknowledge that the approach (i.e., the iterative fit method in Holme and de Viron, 2013) is simple as professor Holme argued, however, it is also not complex to use the normal Morlet wavelet transformation (NMWT) method to recover the target harmonic signals, since the mathematical principle of this method is clear and simple. Consequently, the NMWT method is also called as the “inaction method” (e.g., Cai et al 2018; Wang et al, 2018) due to its simplicity without any inverse transformation process (Liu et al, 2007). For example, as to a harmonic signal $h(t)$ expressed by $h(t) = A_0 \exp(i\omega(t-t_0))$, where $\omega = \frac{2\pi}{T}$. Here, defining the scale factor $a > 0$. There are two useful properties of NMWT in recovering the target signal (i.e., $h(t)$) as following (the proof can be seen in Liu et al, 2007)

$$\text{Property 1 } W_g h(T, b) = h(b), (\forall t = b, a = T)$$

$$\text{Property 2 } \frac{\partial}{\partial a} |W_g h(a, b)| = 0, (\forall a = T)$$

According to these two properties, we can recover the target harmonic signal accurately. The significance of the above two properties has been discussed in the previous works (e.g., Liu et al, 2007; Duan et al, 2015; Duan and Huang, 2019). Furthermore, given that NMWT owns the edge effect as the common wavelet transformation method does, we need to design a strategy to avoid this effect, the purpose of which is to obtain a more accurate result. Additionally, in our opinion,

each method should have its own advantages and disadvantages to analyze the data, therefore, it is very important to adopt a more appropriate method to analyze the LOD data. Given that the currently observed LOD data series is relatively short (i.e., 1962~2019, only ~57yr), while the periods of the target harmonic signals are relatively long (e.g., the 6yr and 8.6yr), especially the prior information of these target signals (e.g., how their amplitudes change: decaying or increasing) are unknown in advance, so it is very important to adopt an appropriate method to accurately recognize these information in advance and quantitatively isolate these target harmonic signals, otherwise these original harmonic signals may be disturbed by an ineffective approach. Considering this point, we suggest that one should make a necessary simulation test to check the reliability of the results from his (or her) method. In our opinion, NMWT is just an appropriate method to analyze the intradecadal variations in LOD and to quantitatively extract the target harmonic signals. Why? Because NMWT method has an obvious advantage that even though we have not any prior information about the target signals (e.g., decaying or increasing) in advance, we still can recognize these time-varying signals and quantitatively recover them in time domain (e.g., Duan et al, 2017; Duan and Huang, 2019).

In addition, Holme and de Viron (2013) analyzed the LOD variations only in the time domain rather than in the frequency domain, and they suggested that only a 6yr oscillation with an almost constant amplitude exists in the LOD intradecadal variations. However, Fourier spectral analysis (see Figure 1b) of the original LOD data (i.e., the black curve in Figure 1a) and the NMWT time-frequency spectrum (see Figure 1c,d) show that the characteristics of the intradecadal variations in LOD does not present a 6yr oscillation alone. Since neither a single 6yr signal with an almost constant amplitude can well characterize the intradecadal variations (i.e., the green curve in Figure 1a) in the time domain, nor the 6yr signal alone can explain the wide energy-spectrum range on the 5~10yr scales presented in Figure 1b.

Figure 1 a shows the LOD variations on the various scales during 1962~2019, where the residual series (i.e., the green curve) mainly reflects the intradecadal variations, which can be seen in Figure 1b; b presents the Fourier analysis results of the LOD data and the intradecadal variations; b also shows that the energy-spectrum range of the LOD variations on 5~10yr band is wide, which shows that a single 6yr oscillation (a sharp peak) cannot explain such wide spectrum; c,d show the NMWT time-frequency spectrum, which further reveal the periodic components existing on the 5~10yr scales.

Secondly, according to the constructive suggestions provided by professor Holme, we will further make some simulation tests to check our method effectiveness and to show the reliability of our results.

Of course, we greatly appreciate the efforts of professor Holme in the research of the LOD intradecadal variations in the time domain. Nevertheless, we would like to prefer this viewpoint that one should analyze the LOD data in both time and frequency domain rather than only in the time-domain. As mentioned above, the frequency domain result of the original LOD variations on the 5~10 yr scales (without involving the other mathematical dealing process, besides a running average to eliminate the seasonal signals) shows a wide energy-spectrum range (i.e., the blue curve in Figure 1b) instead of a single 6yr sharp peak. That is to say, using a single 6yr oscillation with an almost constant amplitude cannot explain this wide energy-spectrum on the intradecadal scales. As Duan et al (2017) indicated, the iterative fitting method (Holme and de Viron 2013) may be not suitable for recognizing and quantitatively extracting the target decaying (or increasing)

harmonic signals from the complex background noises, especially the decreasing (or increasing) information of these harmonic signals is unknown in advance. However, the NMWT method with a high frequency-resolution can recognize the decreasing (or increasing) signals existing in the original observed data and recover them in the time domain perfectly (e.g, Duan et al, 2017; Duan and Huang 2019; also in the supplementary information of this work).

In addition, as mentioned above, in order to avoid the edge effect (EE) of the NMWT method, in this work, we adopt the so-called “BEPME” strategy (how this method works, please see the “Methods” of this work), which is shown to be a very good approach to significantly avoid the EE, and thus we can obtain the more accurate and reliable results (e.g., the target 6yr decaying and 8.6yr increasing signals), otherwise, this EE will significantly disturb the amplitude of these target signals (see the supplementary Figure 2 of this revised manuscript).

On some level, we agree with the idea that detection of the intradecadal variations in LOD might own the “non-unique” issue as suggested by professor Holme. However, in theory, these variations should be rigorously unique, then, why, in practice, the intradecadal variations may have the problem of “non-unique”? The reason of which may come from the different works or methods, which may result in some spectral leakage on the subdecadal frequency band from the low-frequency signals during the fitting process (e.g., removing the background trend), making the intradecadal variations be impure and polluted. However, what we would like to emphasis is that, in this work and the previous works (e.g., Duan et al, 2015; Duan et al, 2017), we used the orthogonal Daubechies wavelet low-pass filter with the 45 order vanishing moment to extract the LOD background trend (with theoretical period $T > 10.67\text{yr}$), although this wavelet filter is not an absolutely ideal filter (see Duan et al, 2017), its frequency spectrum leakage is very small and removing the background trend obtained by this low-pass filter cannot induce a significant influence on the intradecadal variations (we will demonstrate this point in the following analysis). Additionally, this Daubechies wavelet filter has an advantage that it cannot produce the so-called “Gibbs” effect in processing data (e.g., Hu et al, 2005; Duan et al, 2017). Consequently, the residual series (i.e., the green curve) shown in Figure 1a can well characterize the intradecadal variations in LOD, which is also shown in the frequency domain (Figure 1b).

Here, we use the following typical simulation examples to test whether the intradecadal variations in LOD will be disturbed when the background trend (period $T > 10.67\text{yr}$) is removed,

especially, to check whether the existence of the 8.6yr signal in LOD is the consequence of the removal of the background trend? Given that a recent work (Ding 2019) used the AR-z spectrum to analyze the LOD data during a longer time span (i.e., 1760-2018), which shows that there are various periodic components with periods longer than 10yr existing in the LOD data, such as ~149 yr, ~68 yr, ~33 yr, ~22.3 yr, ~18.6 yr, ~13.5 yr, ~11 yr. Assuming that the actual observed LOD variations does own so many signals, here we give the following simulation tests: The data length of this simulation analysis is set to be 686 months (~57yr), which is the same as that of the actual LOD data (i.e., 1962~2019), and the data sampling rate is defined as 1month. Here, the background trend is defined as the variations with period $T > 10$ yr, and the expression of this simulated background trend (“Trend” for short) in the following part is written as

$$\text{Trend}(t)=h_1(t)+h_2(t)+h_3(t)+h_4(t)+h_5(t)+h_6(t)+h_7(t)=\sum_{i=1}^7 h_i(t) \quad (1)$$

where, $t \sim [0:685]$ and the time interval is 1month, $h_i(t)$ expresses the i th signal component, i.e., $h_i(t) = A_i \sin(2\pi f_i t)$; here, f_i signifies the frequency of the i th component, and A_i is the i th amplitude, and we can determine these A_i ($i=1,2,3\dots7$) values according to the relative amplitude values given by Ding (2019).

Furthermore,

$f_1=1/(149 \times 12)$ cpm;	$h_1=0.2\sin(2\pi f_1 t)$;	% 149yr periodic component
$f_2=1/(68 \times 12)$ cpm;	$h_2=0.26\sin(2\pi f_2 t)$;	% 68yr periodic component
$f_3=1/(33 \times 12)$ cpm;	$h_3=\sin(2\pi f_3 t)$;	% 33yr periodic component
$f_4=1/(22.3 \times 12)$ cpm;	$h_4=0.63\sin(2\pi f_4 t)$;	% 22.3yr periodic component
$f_5=1/(18.6 \times 12)$ cpm;	$h_5=0.2\sin(2\pi f_5 t)$;	% 18.6yr periodic component
$f_6=1/(13.5 \times 12)$ cpm;	$h_6=0.05\sin(2\pi f_6 t)$;	% 13.5yr periodic component
$f_7=1/(11 \times 12)$ cpm;	$h_7=0.06\sin(2\pi f_7 t)$;	% 11yr periodic component

where, “cpm” means “cycles-per-month”.

Constructing the following two simulated series

The first series: $Y(t)$ (i.e., Trend+8.6yr term+6yr term+noise term)

$$Y(t)=\text{Trend}+y_1(t)+y_2(t)+\text{noise}(t) \quad (2)$$

where, $y_1(t)$ and $y_2(t)$ are respective the 8.6yr term and 6yr term, noise (t) is a stochastic noise term.

And

$$y_1(t)=0.08\sin(2\pi f_8 t); \quad y_2(t)=0.12\sin(2\pi f_9 t)$$

here, $f_8=1/(8.6 \times 12)$ cpm, $f_9=1/(6 \times 12)$ cpm,

The second series: $H(t)$ (i.e., 8.6yr term +6 yr term)

$$H(t)=y_1(t)+y_2(t) \quad (3)$$

We make the following two types of comparisons

- 1) Comparison of the trend obtained by low-pass filter (the green curve in Figure 2a) and the simulated original background Trend (red curve in Figure 2a);
- 2) Comparison of the residual series obtained by removing the background trend (blue curve in Figure 2b) and the above $H(t)$ (the purple curve in Figure 2b).

Figure 3 shows the comparison of the NMWT time-frequency spectrum of the $H(t)$ signal and the residual series after removing the background trend, while Figure 4 indicates the comparison of the Fourier spectrum between the residual series and $H(t)$. The above simulation results (Figures 2, 3, 4) show that the results after removing the background trend are general well consistent with the original intradecadal variations $H(t)$ (i.e., the 6yr+8.6yr). Of course, according to the above test results (Figure 2, Figure 3 and Figure 4), we cannot guarantee that the residual series obtained in this work must be completely accurate, but we can demonstrate that this result is accurate enough in general, in other words, from which we can obtain the reliable target harmonic signals. In addition, this work actually focuses on the data analysis in both time and frequency domain (Figure 1), which differs from the previous work (Holme and de Viron, 2013). Moreover, the time-frequency domain results (Figure 1b,c,d) show that the LOD intradecadal variations with a wide energy-spectrum range present the different frequency components. Consequently, from the perspective of time-frequency joint analysis, this work indeed can give a more comprehensive result, since this result not only presents a 6yr oscillation, but also shows an 8.6yr signal and its time-varying feature.

Figure 2 a shows the simulated LOD variations; b indicates the comparison of the residual series and the original simulated intradecadal variations.

Figure 3 a,b show the NMWT time-frequency spectrum results of the residual series after removing the background trend; c, d indicate the NMWT time-frequency spectrum results of the simulated $H(t)$ (i.e., 6yr term+8.6yr term).

Figure 4 Comparison of the residual series and the simulated signal (Trend+6yr+8.6yr) in frequency domain.

Conversely, if the intradecadal variations only contain a single 6yr oscillation with an almost constant amplitude as suggested by previous work (Holme and de Viron, 2013), then the Fourier spectrum should present a sharp peak locating at the 6yr period, however the actual LOD intradecadal variations in frequency domain does not support this case (see the blue curve in Figure 1b). That is to say, the actual LOD intradecadal variations should own other frequency components besides the 6yr component, moreover, NMWT spectrum clearly shows the existence of an 8.6yr oscillation.

According to the above discussion, we further revise and improve our original manuscript, please see the revised version, i.e., page 2, lines 50th~56th; page 4, lines 87th-91th; the supplementary information of this revised manuscript, pages 11~14, lines 214th~282th.

Thirdly, as mentioned above, adopting an appropriate mathematical method is important to accurately analyze the LOD intradecadal variations, while the method (i.e., NMWT+BEPME) used in this work is just an appropriate method. In the above simulation tests, the results of which indicate that the influence from removal of the background trend (obtained by wavelet low-pass filter) on the LOD intradecadal variations is small. That is to say, it is reliable to extract the LOD intradecadal variations from the residual series (i.e., the green curve) shown in Figure 1a. Here, we further test whether the target harmonic signals (e.g., 6yr term and 8.6yr term) are disturbed by removing the background trend? Considering the following composite signal

$$Y(t) = \text{Trend}(t) + 8.6\text{yr term} + 6\text{yr term} + \text{noise term}$$

where, the two intradecadal terms (i.e., 6yr and 8.6yr) are simulated respectively through the following three cases:

- 1) Amplitude steady signals, 2) Amplitude decaying signals, 3) Amplitude increasing signals

Simulation test 1: Recovery of the 6yr oscillation in terms of the three cases

In our work, how to recover the above three types of 6yr oscillation signals from the above simulated composite signal- $Y(t)$? In each case, we perform the following steps:

Firstly, we use the wavelet low-pass filter mentioned above to obtain and remove the background trend from $Y(t)$ series;

Secondly, we obtain the residual series;

Finally, we use the NMWT+BEPME method to recover the target 6yr signal from the

residual series, and the results are shown in Figure 5.

Figure 5 Comparison of the the 6yr signals (blue curves) recovered by the NMWT+BEPME method and the original simulated 6yr signals (red curves) in three cases; here, the strategy of avoiding the NMWT edge effect is so-called “BEPME” approach illustrated in the manuscript.

Simulation test 2: Recovery of the 8.6yr oscillation in terms of the three cases

How to recover the three types of the 8.6yr oscillation signals from the above simulated composite signal- $Y(t)$? the data processing steps are the same as that of the 6yr oscillation recovery mentioned above. And the results are shown in Figure 6.

Figure 6 Comparison of the the 8.6yr signals (blue curves) recovered by the NMWT+BEPME method and the original simulated 8.6yr signals (red curves) in three cases; here, the strategy of avoiding the NMWT edge effect is so-called “BEPME” approach illustrated in the manuscript.

According to the above simulated tests, the results of the 6yr and 8.6yr signals (see Figure 5 and Figure 6) recovered by this method (i.e., NMWT+BEPME) are general well consistent with the original simulated signals, although they are not the absolutely accurate results, especially at the maximum side of the data, where the amplitudes of the target harmonic signals are not recovered perfectly, i.e., the recovered results are always somewhat smaller than the real values,

which shows that the process of removing the background trend may bring some influences on the recovery of the target signals, and this point has been discussed in this manuscript. Nevertheless, the target 6yr and 8.6yr signals in the LOD are recovered perfectly in general. In summary, according to the simulation tests, we argue that the results of the intradecadal variations (including 6yr and 8.6yr) from our method (i.e., NMWT+BEPME) are “unique” rather than the highly “non-unique”. Even though the “non-unique” issue does exist, which should only belong to the “local accuracy” issue, but it cannot influence the overall quality of the data results.

In addition, our method (NMWT+BEPME) does not need to set the 6 fitting parameters in advance, but this method firstly recognizes the target time-varying (e.g., the decaying or increasing) harmonic signals, and then recovers them in the time-domain with a high accuracy from the original composite mix-frequency signals. That is to say, the data processing strategy of using our method is to recognize and extract the target harmonic signals in time domain, while this recovered result is just the expected final result. Of course, it depends on one’s needs to determine whether (or not) to fit these recovered results from the method (NMWT+BEPME) to obtain the relevant parameters (e.g., decaying factor), the process of which also can be seen in our recent work (Duan et al, 2017; Duan and Huang, 2019). This is the reason why we would like to argue that the method used in this work is a more appropriate method to analyze the original LOD data, since we do not need to know the prior information about the amplitude changes of the 6yr and 8.6 yr signals in advance, but we still can recognize them and recover them in the time domain.

According to the above discussion, we have revised the original manuscript, please see the revised version, page 2, lines 50th~56th; pages 2~3, lines 58th~61th; page 3, lines 71th~75th; page 4, lines 87th~92th; page 6, lines 137th~140th; in the supplementary information of this work, please see the pages 11~15, lines 217th~319th.

What is the necessity of the two different periods? This relates to the removal of a slowly varying trend. This trend has sufficient variation that it can co-vary with the harmonic components solved for here. Thus the problem, even as established, is seriously non-unique. In the original paper, I iterated between the long-term trend and the oscillation – I removed the oscillation before detrending again, and by doing this got a much better fit for a single constant amplitude component. This is not done here – the problem can be seen in 1992 where the background trend

shows a peak at exactly the same time as the 6 year oscillation.

The iterative approach allowed this variation to be adjusted by the background trend, the approach in this paper requires a low value of the oscillation at this time, which is where the mix of the 6 year and 8 year periods comes in – at this point they are out of phase, and so the background trend does not need to be altered. To avoid this cancellation happening at other times, the growing and decaying exponentials mean that there is not cancellation apart from at this one time.

Response: Thank professor Richard Holme for his careful and constructive comments. We agree with this opinion that removal of the background trend may influence the intradecadal variations, nevertheless, from the above simulation analysis, this influence involved in this work is proved to be small, especially, the two obviously different periods (i.e., 6 yr and 8.6yr) are not the consequence of the removal of the varying trend, but the intradecadal LOD variations contain these two obviously periodic components. Importantly and obviously, the Fourier spectral results of the original LOD data (i.e., the blue curve shown in Figure 1b) still shows a wide energy-spectrum range on the 5~10yr band, which cannot be explained by a single 6yr peak.

Additionally, as mentioned above, one not only should analyze the data in the time domain, but also should consider the issue in frequency domain. Because, only in the time domain to watch the data, the superposition of the different signals with different frequencies will present the so-called “co-vary” behavior mentioned by professor Holme, and it is difficult to distinguish them only in the time domain. However, these different signals with different frequencies can be easily separated each other in the frequency domain. In fact, the issue that “the problem can be seen in 1992 where the background trend shows a peak at exactly the same time as the 6 year oscillation” mentioned by professor Holme can be solved by the time-frequency analysis. As Holme mentioned, “1992 where the background trend shows a peak”, this feature actually reflects the decadal variations (i.e., the background trend) with the period $T > 10.67$ yr, while the residual series represent the intradecadal variations on the 5~10yr band. Hence, the coincidence between the background trend and the intradecadal variations during 1990s does not show the “co-vary” between the background trend and the 6yr, but actually represents the changes of the background trend, because, at this time, the intradecadal variations nearly “disappear” due to the superposition of all the components (at least the 6yr term and the 8.6 yr term) existing on the intradecadal scales.

The results of time-frequency analysis in this work show that there are two principal periodic components (i.e., the 6yr and 8.6yr terms) existing in the LOD intradecadal variations. The frequency resolution of NMWT method is high enough and it can easily distinguish these two different frequency components. Furthermore, the method used in this work does not need such a following requirement (mentioned by the above comments), i.e., “require a low value of the oscillation at this time, which is where the mix of the 6 year and 8 year periods comes in – at this point they are out of phase, and so the background trend does not need to be altered”, this low value occurred during 1990s is actually the consequence of the superposition of all the subdecadal components (e.g., at least the 6yr and 8.6y oscillations) existing in LOD variations, which essentially presents the varying characteristic of the intradecadal variations in the time domain.

According to the above comments and the discussion, we have further revised the manuscript, please see the revised version, page 3, lines 71th~75th.

Both papers fit the data, and therefore on some level it is not possible to argue that one is better than the other. However, the simplicity of the first paper (a smooth curve combined with a two parameter fit) compared with complexity of the second (highly developed spectral methods for which the time series may be too short, with a 6 parameter fit that fits the data less well) suggests to my mind that this work does not provide an advance in our understanding. Note also that the fit is not perfect in either case – this is because the signal being fit is not stationary, and shows jumps (probably due to physical processes) – again see figure S3. So having arguments about including higher complexity of statistically stationary elements is a bit futile. This argument would need a discussion of how great an improvement in fit the more complex model provides, including allowing time variation of the background signal – and I would argue that instead the fit is worse.

Response: Thank professor Richard Holme for his comments. As mentioned above, we greatly acknowledge the efforts of professor Holme on the fitting of the intradecadal variations in LOD. However, we believe that our results from the method (i.e., NMWT+BEPME) used in this work are reliable, the reasons of which are listed as following:

First This work is a more comprehensive work relative to the previous works, since this work analyzes the LOD data in both time and frequency domains. Importantly, we can demonstrate that the results from our method are reliable through the relevant typical simulation tests.

Second The primary criteria to judge whether an analysis method is good (or not) should depend on whether this method is effective (or not) to accurately recognize and extract the target harmonic signals existing in LOD variations rather than the simplicity (or not) of this method. Besides, as mentioned above, it is also not complicate for the NMWT method to extract the target harmonic signals.

Here, judging whether an effective method (that we mentioned above) is effective or not should be that this method can achieve the following aim: this method not only has the ability to recognize the target harmonic signals with different periods and time-varying information (e.g., decaying or increasing), but also this method is able to extract these time-domain signals from the original observed data with complex background noise. While NMWT method can achieve the above aim nicely.

Furthermore, we used the NMWT method to recognize the two different periodic components (e.g., 6yr and 8.6yr) existing in LOD variations, especially we isolate the 8.6yr oscillation in time domain and find its increasing phenomenon, and further find its close relations to the geomagnetic jerks. Therefore, this work provides an advance in our understanding of the LOD variations in nature relative to the previous work. Through the simulation analysis, although the results from this work are not necessarily exact outcomes with the 100% accuracy in detail (this point has been discussed in the revised manuscript, see page 6, lines 136th~145th), the overall outcomes that we mainly focus on are accurate enough. Since our results and the method used in this work can stand the typical simulation tests, hence, these overall results of this work including the main conclusions are reliable.

The big problem is the apparent belief that complicated modelling methods (such as wavelets) will solve this problem. Even simple analysis of splitting different frequencies of signals is not robust – it is highly likely that there are components of intervening frequencies in both the background signal and the oscillation. The methods rely on a (non-stationary) series of (noisy) data, with insufficient length to justify complex analysis – and yet the final fit is not as good as the simpler analysis. There is one possible test – I note in figure 3, the data end at 2015, so there should be another 4 years of data by now to provide a test of just how good a predictive model is provided by the two models.

Response: Thank professor Holme for providing the comments. According to the above tests, as professor Holme suggested, it is indeed likely that there are components of intervening frequencies in both the background signal and the intradecadal oscillation (or the residual series), which is due to the disturbance from the relevant filtering process. Nevertheless, such influence in this work is proved by the simulation tests to be very small in general, it cannot influence the overall quality of the target signal results. That is to say, the results (i.e., the 8.6yr and 6yr oscillations in time domain) recovered by this work are reliable. In summary, the method (NMWT+BEPME) used in this work is indeed suitable for analyzing the current LOD data. Importantly, the BEPME strategy designed in this work can effectively eliminate the edge effect of the NMWT method to help the NMWT method to get a better and more reliable result (please see the Method of this work).

In addition, as to the question why in Figure 3, the data end at 2015 in the original manuscript, which is due to the requirement from the BEPME strategy, here we need to correct it as the data end at 2016 in the revised manuscript. For example, Figure 7 shows that we only need to search for the local extreme point times (t_i and t_j) locating nearest to the boundaries, and the data outside the time range of extreme points (i.e., $t < t_i$ and $t > t_j$) are removed. Consequently, when we use BEPME strategy to process the 6yr oscillation, we find this extreme point time locates at ~ 2016.4 (see Figure 8a), while the 8.6yr oscillation locates at ~ 2016.0 (see Figure 8b), so we only use the data end at 2016.0, please see the Figure 3 in the revised manuscript.

According to the above comments, we have further revised the manuscript, please see the revised version, page5, lines 117th~119th.

Figure 7 On the boundary extreme point times (t_i and t_j) at the two sides of the BEPME strategy.

Figure 8. On the right boundary extreme points (t_i and t_j) of the 6yr signal and 8.6yr signal recovered by NMWT+BEPME method.

The discussion of the physical mechanisms does not bring anything new, or anything robust. The problem with such a speculative analysis, as I have found to my cost, is that even if qualified (not really done here), once it enters the literature, other workers seem to believe it proven and then chase off in unjustified directions of work. There is reference to work of Gillet – again this relies on the validity of the transform methods and a clear separation between short and long period signal, so at least it is good that the two analyses are in agreement. However, I note that the flow analysis that Gillet performs ended up with just a 6 year variation – in fairness, long before I got involved in this story.

Response: Thank professor Holme for comments. Here, we greatly acknowledge the pioneering works (e.g., Holme and de Viron, 2013; Gillet et al, 2010) in this research field. Nevertheless, this work differs from the previous works (e.g., Holme and de Viron, 2013; Gillet et al, 2010), which mainly focused on the 6yr oscillation alone. This work actually not only shows the 6yr oscillation existing in the time-frequency spectrum, but also presents an 8.6yr periodic signal, meanwhile, we find a new phenomena (i.e., the 8.6-year signal amplitude increasing, a good correspondence between this extremes of this 8.6yr signal and the geomagnetic jerk timings). Besides, we focus on combining the findings of this work with the previous works (e.g., Holme and de Viron, 2005; Holme and de Viron, 2013; Chulliat et al, 2015; Gillet et al, 2015;2019; Aubert and Finlay, 2019) to give a reasonable explanation of these new phenomena. Through the typical simulation tests, the results in this work are showed to be reliable and robust. Consequently, this work is innovative,

and it is not speculative, but well founded and reasonable.

In summary: Thank professor Holme very much once again for providing the helpful and constructive comments. We made the corresponding typical numerical simulations, which confirms that NMWT method is a powerful and effective mathematical tool in analyzing current LOD data and that the recovered target harmonic oscillation are reliable. In our opinion, for a time series consisting of a varying trend and a stationary periodic signal, this iterative fit method (Holme and de Viron, 2013) is effective to recover this trend and the target steady harmonic oscillation, while for analysis of the time-varying (e.g., the decaying or increasing in amplitude) signals, especially the prior information of these signals are unknown in advance, this iterative fit method is not an ideal approach. Then, how to accurately recognize these signals and further quantitatively recover them, especially when the observed series is not too long while the periods of the target signals are relatively long (e.g., the intradecadal variations of the current LOD observations from 1962 to now) ? The NMWT+BEPME method adopted in this work is just a good answer.

Minor points –

Reference 6 talks about intradecadal variation, but does not identify the 6 year period as clearly separated. This reference needs to appear, but not at the point in the text where it now does.

Response: Thank professor Richard Holme for his comments. According to these comments, we revised the manuscript, please see the revised version, i.e., the pages 2, lines 31th~33th; pages 2~3, lines 58th~61th.

What AAM series is used? Is there any smoothing of the data after this removal (a running average or a low pass filter?)

Response: Thank professor Richard Holme for his comments. We are very sorry that we have not explained the LOD data clearly in our original manuscript. The AAM series used in this work is loaded from the IERS website. Yes, there is smoothing of the data after this removal, the smoothing method adopted in this work is to apply the 12-month and 6-month running average to eliminate the corresponding remaining seasonal signals (i.e., the annual and semi-annual terms).

According to these comments, we have further revised the manuscript, please see the revised version, i.e., page 2, lines 48th~50th.

References

- Aubert, J., Finlay, C.C. Geomagnetic jerks and rapid hydromagnetic waves focusing at Earth's core surface. *Nature Geoscience*, 12, 393-398 (2019).
- Chulliat, A., Alken, P., Maus, S. Fast equatorial waves propagating at the top of the Earth's core. *Geophys. Res. Lett.*, 42,3321-3329 (2015).
- Cai, S., Liu, L. T., Wang, G.C., 2018., Short-term tidal level prediction using normal time-frequency transform. *Ocean Engineering.*, 156, 489-499.
- Duan, P.S., Liu, G.Y., Liu, L.T., Hu, X.G., Hao, X.G., Huang, Y., Zhang, Z.M., Wang, B.B. Recovery of the 6 year signal in length of day and its long term decreasing trend. *Earth Planets Space*. 67:161, doi:10.1186/s 40623-015-0328-6 (2015).
- Duan, P.S., Liu, G.Y, Hu, X.G, Sun, Y.F, Li, H.L. Possible damping model of the 6 year oscillation signal in length of day. *Phys. Earth Planet. Inter*, 265. 35-42 (2017).
- Duan P S, Liu G Y, Hu X G, Zhao J, Huang C L. Mechanism of the interannual oscillation in length of day and its constraints on the electromagnetic coupling at the core-mantle boundary. *Earth Planet. Sci. Lett.* 482, 245-252 (2018).
- Duan, P. S and Huang, C. L. Application of normal Morlet wavelet transform method to the damped harmonic analysis: On the isolation of the seismic normal modes (${}_0S_0$ and ${}_0S_5$) in time domain. *Phys. Earth Planet. Inter*, 288, 26-36 (2019).
- Ding, H. Attenuation and excitation of the ~6 year oscillation in the length-of-day variation. *Earth Planet. Sci. Lett.*, 507,131-139 (2019).
- Holme, R., de Viron, O..Characterization and implications of intradecadal variations in length of day. *Nature*. 499, 202-205 (2013).
- Holme R, Viron O. Geomagnetic jerks and a high-resolution length-of-day profile for core studies. *Geophys J Int* 160,435-439 (2005).
- Hu, X. G, L.T. Liu, J. Hinderer, and H. P. Sun (2005), Wavelet filter analysis of local atmospheric pressure effects on gravity variations, *J. Geod.*, 79:447-459.

Liu, L.T., Hsu, H.T., Grafarend, E.W. Normal Morlet wavelet transform and its application to the Earth's polar motion. *J. Geophys. Res.* 112, B08401,doi10.1029/2006JB004895 (2007).

Wang, G.C., Liu, L.T., Xu, A.G., Pan, F., Cai, Z.W., Xiao, S.H., Tu, Y., Li, Z.H. On the capabilities of the inaction method for extracting the periodic components from GPS clock data. *GPS. Solution*, 22:92 (2018).

REVIEWERS' COMMENTS:

Reviewer #1 (Remarks to the Author):

In my initial review, I gave 2 major points I believed should be addressed, namely,

1. A failure to discuss the uncertainty of jerk timings
2. A failure to adequately demonstrate
 - a. how the decadal variation of LOD series was derived
 - b. that TMWT cannot resolve periods as close as 6yr and 8.5yr

From the revised manuscript and supplement, I can now see that both these points have been expanded upon.

On my first point, I think the reader is now given enough background to see that there are various views expressed in the literature and the authors here support one of these.

On my second point, I think the manuscript coupled with the supplement provides the information required.

I see that the concerns of the other referees have been responded to, while I don't see any obvious error in the response (and am generally happy to see synthetic tests to back up the authors arguments) it would perhaps be better for the other referees to affirm to this.

There are a handful of references such as "see supplement" that should refer explicitly to where in the supplement a reader should look, especially since this supplement has grown somewhat.

From my point of view the content of the manuscript is now acceptable to publish, though the sections of text that have been added could use a copy edit as the grammar is awkward in several places.

Reviewer #2 (Remarks to the Author):

Please find my report on on the revised manuscript entitled « Intradecadal variations in length-of-day and their correspondence with geomagnetic jerks » by P. Duan and C. Huang, and submitted to Nature Communications.

The authors have performed important efforts to test the robustness of their results. This point should be acknowledged. They have also significantly changed their discussion of the geophysical implications.

Provided they take into account my comments below, I consider the manuscript is now acceptable for publication with minor revisions.

In particular I have to important remarks, detailed below : the authors

- should modify their discussion of the inner core oscillations, which cannot be considered without motions in the fluid outer core ;
- should avoid mentioning that jerks imply LOD changes, since they may well result from the same cause (concomittance is not causality !)

see attached document for detailed comments.

Reports on the revised manuscript entitled « Intradecadal variations in length-of-day and their correspondence with geomagnetic jerks » by P. Duan and C. Huang, and submitted to Nature Communications.

The authors have performed important efforts to test the robustness of their results. This point should be acknowledged. They have also significantly changed their discussion of the geophysical implications. Provided they take into account my comments below, I consider the manuscript is now acceptable for publication with minor revisions. In particular I have to important remarks, detailed below : the authors

- should modify their discussion of the inner core oscillations, which cannot be considered without motions in the fluid outer core ;

- should avoid mentioning that jerks imply LOD changes, since they may well result from the same cause (concomittance is not causality !)

list of comments :

- l 17 : « indicates that the geomagnetic jerks may excite the 8.6-year signal amplitude ». I do not agree on this, they may as well have a similar origin (see Aubert & Finlay, 2019)

- l 160-161 : « Recently, the idea that the geomagnetic jerks originating from the Earth interiors has been widely accepted »... this idea is accepted since the 1980's !!

- l 162 : rather than refer to Ref. 30 (Bloxham et al, 2002), I would refer to more complete time-dependent core flow inversions (e.g. Jackson et al 1991 ; Jackson et al 1997)

- l 164 : « Therefore, the jerk events may excite the LOD variations on the intradecadal scales... » I do not agree, they may result from a same source !! (see Aubert & Finlay, 2019)

- l 207 : « From the intradecadal variations in LOD and their time-varying characteristics, one can infer the geomagnetic field strength inside the Earth's core » why refer to Ref. 18 there ?

- On the same page : If I follow the scenario by the authors, torsional waves may only correspond to the 6 yr signal, the 8.6 yr signal, being related to another physical phenomenon (trapped waves in a stratified layer or QG-Alfven waves as in the study of Aubert & Finlay (2019). Depending on the physics chosen, the link to the magnetic field within the core will differ. This should be clearly stated somewhere.

- l 23 and below (+ rebuttal on pages 32-33) : I agree with the derivation and the 0.2 deg/yr for the inner core... if no angular momentum is carried by the fluid outer core. However, the issue here is that it corresponds to a ~4.6 km/yr rms zonal velocity at the inner core equator, which is about 10 times what is inferred from geomagnetic field changes !! this is one reason for accounting for the fluid core even in the case of a MICG. Another reason is the EM torque between the inner core and the fluid outer core, which will couple the two. For these reasons, one cannot explain LOD changes with angular momentum only in the inner core... (it is not possible to ignore the outer core in this ballance).

- l 231 : « is likely due »... is possibly due

- l 237-238 : « If it is this case, then it will be not appropriate to use the observed 6-year period to estimate the cylindrical radial component of the magnetic field »... unless if 6-yr were to correspond to a harmonic of the 8.6 yr signal, which would still be possible given the uneven relation between eigen frequencies of the torsional modes, in the likely case where the torsional Alfven wave velocity is not constant throughout the outer core (see Gillet et al, 2017).

- on page 10 : another possibility is QG-Alfven waves as supported by Aubert & Fourier 2019... but why would their signature be necessarily periodic ?

- l 272-273 : « ... that geomagnetic jerks are probably responsible for the increasing of the amplitude » : see above points for l 17 and 164.

- « ... these jerks can induce the acceleration of the azimuthal flow motions »... can be associated with the acceleration of the azimuthal flow motions

- « ... which makes the amplitude of the 8.6yr signal continuously increasing » at some point it should decrease. Why do the authors want to see there a causality and not just a concomitance ?

- l 291 : « while this work further provides a directly observed evidence to show this oscillatory behavior »... SA pulses (e.g. Finlay et al, 2016) seem to occur every 3 yr or so, making jerks periodicity ~6 yr. This differs from the 8.6 yr periodicity argued for by the authors. How would the authors reconcile this ?

Reviewer #3 (Remarks to the Author):

Referee: Richard Holme

The authors have certainly done a very substantial amount of work. I worry that much of the work doesn't answer the question - they do demonstrate from the synthetics that they are able to recover their proposed double harmonic signal, with exponential growth and decay. What they do not demonstrate is that this signal would could not be explained in another way (as for example a trend and one harmonic signal).

The authors argue strongly that their approach is more complete because they consider both the time and frequency domain. However, considering the frequency domain is not an automatic solution to problems. In the end, all they can do is fit data with some weighting applied - a fourier method does this ias much as a parameterised method, it is just hidden by formalism. If I have one major worry with the work, is that the authors believe more refined mathematical methods will give "truth" - I doubt this strongly.

However, as I see that the other reviewers were more friendly, I cannot argue against the paper. I make one request, probably in the supplemental information - the authors provide in one place the detailed parameterisation of their fit (frequencies and exponential rates). This will allow testing of their model against future data (some of this is already available) and also past data from occultation information. This can then be tested against a single wave fit (which anyway breaks down sometime back in time in the 1950s) to allow, in a couple of years, an objective test as to whether their model has any long term predictive properties.

Response to reviewers

Reviewer #1 (Remarks to the Author):

In my initial review, I gave 2 major points I believed should be addressed, namely,

1. A failure to discuss the uncertainty of jerk timings
2. A failure to adequately demonstrate
 - a. how the decadal variation of LOD series was derived
 - b. that TMWT cannot resolve periods as close as 6yr and 8.5yr

From the revised manuscript and supplement, I can now see that both these points have been expanded upon.

On my first point, I think the reader is now given enough background to see that there are various views expressed in the literature and the authors here support one of these.

On my second point, I think the manuscript coupled with the supplement provides the information required.

I see that the concerns of the other referees have been responded to, while I don't see any obvious error in the response (and am generally happy to see synthetic tests to back up the authors arguments) it would perhaps be better for the other referees to affirm to this.

There are a handful of references such as "see supplement" that should refer explicitly to where in the supplement a reader should look, especially since this supplement has grown somewhat.

From my point of view the content of the manuscript is now acceptable to publish, though the sections of text that have been added could use a copy edit as the grammar is awkward in several places.

Response: Thank the reviewer very much for the good comments. According to these suggestions, we have referred explicitly the relevant references such as "see SI" in the revised manuscript, for example, please see page 2, line 34th; page 3, line 76th; page 5, line 122th.

In addition, we have further checked the manuscript and revised the corresponding grammar and English language. Thank the reviewer very much once again.

Reviewer #2 (Remarks to the Author):

The authors have performed important efforts to test the robustness of their results. This point should be acknowledged. They have also significantly changed their discussion of the geophysical implications. Provided they take into account my comments below, I consider the manuscript is now acceptable for publication with minor revisions.

Response: Thank the reviewer very much for the careful comments and good suggestions. Here, according to these suggestions, we have further revised the manuscript, the detailed response information of which can be seen in the following part.

In particular I have to important remarks, detailed below :

the authors

-should modify their discussion of the inner core oscillations, which cannot be considered without motions in the fluid outer core;

Response: Thank the reviewer for the good suggestion. We agree with the viewpoint that the inner core oscillation (under the gravitational torque from the mantle) will necessarily involve the fluid core coupling, which is just consistent with our recent published work (Duan & Huang, 2020) that focused on the inner-outer core coupling issue under the action of the electromagnetic coupling effects. According to the reviewer's suggestion, we have modified the manuscript, please see the modified version: page 2, lines 45th-46th; page 9, lines 249th-251th.

-should avoid mentioning that jerks imply LOD changes, since they may well result from the same cause (concomittance is not causality !)

Response: According to a recent simulation work (Aubert & Finlay, 2019) about geomagnetic jerk origin, we agree with this suggestion that geomagnetic jerks and the 8.6yr signal in LOD may result from a same physical cause. According to this suggestion, we have further revised the manuscript, please see the revised version: page 7, lines 205th-207th; page 14, lines, 383th-388th.

list of comments :

- l 17 : « indicates that the geomagnetic jerks may excite the 8.6-year signal amplitude ». I do not agree on this, they may as well have a similar origin (see Aubert & Finlay, 2019)

Response: Thank the reviewer for the good suggestion. As mentioned above, we have revised the original Abstract of our manuscript, please see the ‘Abstract part’ in the revised version, i.e., page 1, lines 10th-17th.

- l 160-161 : « Recently, the idea that the geomagnetic jerks originating from the Earth interiors has been widely accepted »... this idea is accepted since the 1980’s !!

Response: Thank the reviewer for nice comment. In the revised manuscript, we have removed the word of “Recently”, please see the revised version, i.e., page 5, line 142th.

- l 162 : rather than refer to Ref. 30 (Bloxham et al, 2002), I would refer to more complete time dependent core flow inversions (e.g. Jackson et al 1991 ; Jackson et al 1997)

Response: Thank the reviewer for providing the nice comment. According to this advice, we have further revised the manuscript and cited this work (e.g., Jackson et al 1997) in the revised version, please see page 5, line, 144th.

- l 164 : « Therefore, the jerk events may excite the LOD variations on the intradecadal scales... » I do not agree, they may result from a same source !! (see Aubert & Finlay, 2019)

Response: As mentioned above, we have revised the original point, please see the revised version, i.e., page 5, line 146th.

- l 207 : « From the intradecadal variations in LOD and their time-varying characteristics, one can infer the geomagnetic field strength inside the Earth’s core » why refer to Ref. 18 there ?

Response: Thank the reviewer for good suggestion. In the revised manuscript, we have removed the reference 18, please see the revised version, i.e., page 7, line 183th.

- On the same page : If I follow the scenario by the authors, torsional waves may only correspond to the 6 yr signal, the 8.6 yr signal, being related to another physical phenomenon (trapped waves in a stratified layer or QG-Alfven waves as in the study of Aubert & Finlay (2019). Depending on the physics chosen, the link to the magnetic field within the core will differ. This should be clearly stated somewhere.

Response: Thank the reviewer's comment. This is a good suggestion, since this statement suggested by the reviewer can make the relevant conclusion of this work be more explicit. Consequently, we have added this discussion in our revised manuscript, i.e., "Here an additional point is worthy of further discussion, i.e., if the 8.6-year signal origin is really attributed to the trapped waves in a stratified layer at the Earth' core surface or the so-called QG-Alfvén waves (Aubert & Finlay, 2019), then the fast torsional waves detected by Gillet et al(2010) will only correspond to the 6-year oscillation. Consequently, depending on the physics chosen, the link to the magnetic field within the FOC will differ." Please see the revised version: page 9, lines 260th-264th.

- 1 23 and below (+ rebuttal on pages 32-33) : I agree with the derivation and the 0.2 deg/yr for the inner core... if no angular momentum is carried by the fluid outer core. However, the issue here is that it corresponds to a ~4.6 km/yr rms zonal velocity at the inner core equator, which is about 10 times what is inferred from geomagnetic field changes !! this is one reason for accounting for the fluid core even in the case of a MICG. Another reason is the EM torque between the inner core and the fluid outer core, which will couple the two. For these reasons, one cannot explain LOD changes with angular momentum only in the inner core... (it is not possible to ignore the outer core in this ballance).

Response: Thank the reviewer very much for the valuable suggestion. As mentioned above, we agree with that the outer core coupling motions is accompanied by the inner core oscillation (see Duan & Huang, 2020). In the revised version, we have further added the above suggestion to our revised manuscript, please see page 7, lines 205th-207th; page 14, lines 383th-388th.

- 1 231 : « is likely due »... is possibly due

Response: According to this suggestion, we have revised the manuscript. Please see the revised version, i.e., page 8, line 288.

- 1 237-238 : « If it is this case, then it will be not appropriate to use the observed 6-year period to

estimate the cylindrical radial component of the magnetic field »... unless if 6-yr were to correspond to a harmonic of the 8.6 yr signal, which would still be possible given the uneven relation between eigen-frequencies of the torsional modes, in the likely case where the torsional Alfvén wave velocity is not constant throughout the outer core (see Gillet et al, 2017).

Response: Thank the reviewer very much for the good comments. According to the previous works (e.g., Gillet et al, 2010; Gillet et al, 2015), the 8.6yr signal is possibly attributed to the torsional waves within the fluid outer core, and this work have discussed this case, in which, the period of the torsional waves may be prolonged when these waves propagate from the inner core to the core-mantle boundary equator, since the magnetic field at the Earth's core surface is smaller than at the ICB, so the velocity of these waves are smaller relative to that at the inner core boundary (ICB) as the reviewer suggested. Of course, as the reviewer suggested, this 8.6yr signal is also possible a trapped waves at the Earth's core surface. In the revised work, we also discuss this latter likely case, please see the revised version, i.e., page 9, lines 249th-253th.

- on page 10 : another possibility is QG-Alfvén waves as supported by Aubert & Fourier 2019... but why would their signature be necessarily periodic ?

Response: Thank the reviewer very much for the good question. Given that Chulliat et al (2015) detected an obviously equatorial waves (i.e., the so-called QG-Alfvén waves as supported by Aubert & Fourier 2019) at the surface of the Earth' core, the period of which was found to be ~8.5yr (see Chulliat et al, 2015). Therefore, this signal may be a normal mode signal within the fluid core at the Earth's core surface, so the relevant signature may perform a periodic behavior (with high probability).

- l 272-273 : « ... that geomagnetic jerks are probably responsible for the increasing of the amplitude » : see above points for l 17 and 164.

Response: Thank the reviewer for suggestion. As mentioned above, we have revised the manuscript.

- « ... these jerks can induce the acceleration of the azimuthal flow motions »... can be associated with the acceleration of the azimuthal flow motions

Response: According to this suggestion, we have revised the manuscript, please see page 9, lines 255th-256th.

- « ... which makes the amplitude of the 8.6yr signal continuously increasing » at some point it should decrease. Why do the authors want to see there a causality and not just a concomitance ?

Response: Thank the reviewer very much for the good question. As mentioned above, we agree with that the 8.6yr signal and the jerks may result from the same physical resource. In the revised version, we have revised the relevant statements.

- l 291 : « while this work further provides a directly observed evidence to show this oscillatory behavior »... SA pulses (e.g. Finlay et al, 2016) seem to occur every 3 yr or so, making jerks periodicity ~6 yr. This differs from the 8.6 yr periodicity argued for by the authors. How would the authors reconcile this ?

Response: Thank the reviewer very much for the good question. This work shows that there is a very good correspondence between the 8.6-year signal in length-of-day and geomagnetic jerks, where the geomagnetic jerk data during the past several decades has been recorded by previous works (e.g., Holme & de Viron, 2013; Torta et al, 2014). This finding reveals that the jerk occurrence owns a ~4-year periodic oscillatory behavior, which differs from the previous work (e.g. Finlay et al, 2016) that SA pulses seem to occur every 3 yr or so. In our opinion, this difference is significant to further understand the periodicity of the SA. In addition, this point of jerk occurrence with ~4yr period is also compatible with the other works (e.g., Torta et al, 2014; Kotzé, 2017) which showed that the jerk occurrence presents 3~5yr periodicity over the last several decades.

References

- Aubert, J. & Finlay, C.C. Geomagnetic jerks and rapid hydromagnetic waves focusing at Earth's core surface. *Nature Geoscience*, 12, 393-398 (2019).
- Chulliat, A., Alken, P. & Maus, S. Fast equatorial waves propagating at the top of the Earth's core. *Geophys. Res. Lett.*, 42, 3321-3329 (2015).
- Duan, P. S. & Huang, C. L. On the mantle-inner core gravitational oscillation under the action of the

electromagnetic coupling effects. *J. Geophys. Res: Solid Earth*, 125, e2019JB018863.<https://doi.org/10.1029/2019JB018863> (2020).

Gillet, N., Jault, D., Canet, E. & Fournier, A. Fast torsional waves and strong magnetic field within the Earth's core. *Nature*. 456, 74-77 (2010).

Gillet, N., Jault, D. & Finlay, C. C. Planetary gyre, time-dependent eddies, torsional waves and Equatorial jets at the Earth's core surface, *J. Geophys. Res: Solid Earth*. 120, 3991-4013, doi:10.1002/2014JB011786 (2015).

Holme, R. & de Viron, O. Characterization and implications of intradecadal variations in length of day. *Nature*. 499, 202-205 (2013).

Jackson, A. Time-dependency of tangentially geostrophic core surface motions. *Phys. Earth Planet Inter.*, 103 (3-4), 293-311 (1997).

Kotzé, P.B. The 2014 geomagnetic jerk as observed by southern African magnetic observations. *Earth Planets Space*. 69:17, doi:10.1186/s40623-017-0605-7. (2017)

Torta, J.M., F. J. Pavón-Carrasco, S. Marsal. & Finlay, C. C. Evidence for a new geomagnetic jerk in 2014, *Geophys. Res. Lett.*, 42, 7933–7940, doi:10.1002/2015GL065501 (2015).

Reviewer #3 (Remarks to the Author):

Referee: Richard Holme

The authors have certainly done a very substantial amount of work. I worry that much of the work doesn't answer the question - they do demonstrate from the synthetics that they are able to recover their proposed double harmonic signal, with exponential growth and decay. What they do not demonstrate is that this signal would could not be explained in another way (as for example a trend and one harmonic signal).

The authors argue strongly that their approach is more complete because they consider both the time and frequency domain. However, considering the frequency domain is not an automatic solution to problems. In the end, all they can do is fit data with some weighting applied - a fourier method does this as much as a parameterised method, it is just hidden by formalism. If I have one major worry with the work, is that the authors believe more refined mathematical methods will give "truth" - I doubt this strongly.

However, as I see that the other reviewers were more friendly, I cannot argue against the paper. I

make one request, probably in the supplemental information - the authors provide in one place the detailed parameterisation of their fit (frequencies and exponential rates). This will allow testing of their model against future data (some of this is already available) and also past data from occultation information. This can then be tested against a single wave fit (which anyway breaks down sometime back in time in the 1950s) to allow, in a couple of years, an objective test as to whether their model has any long term predictive properties.

Response: Thank professor Richard Holme for providing the careful and constructive comments. Here, according to the above request, we further provide the detailed parameterization of fit (frequencies and exponential rates) as following:

As to the 6-year signal: the frequency $f \sim 0.0138$ cpm (cycles-per-month) and exponentially decaying rate $\beta \sim 8.4 \times 10^{-4}$ /month, which have been given by our previous work (see Duan et al, 2017), so in this work we directly cite these results, please see the revised manuscript, i.e., page 4, lines 100th-101th; page 20, lines 522th-523th in the caption of Fig.2.

Furthermore, in this work, we first fit the currently recovered 8.6yr signal in LOD using an exponentially increasing model as following (see Fig.4)

$$y(t) = A_0 \exp[\alpha(t-t_0)] \cos[2\pi f(t-t_0)]$$

where, the initial time t_0 is set to be at June, 1982; A_0 refers to the initial amplitude; α expresses the exponentially increasing rate; f signifies the harmonic signal frequency.

The relevant values of these parameters are estimated as following:

$$A_0 \sim 0.06\text{ms}; \alpha \sim +0.00131/\text{month}; f \sim 0.00969\text{cpm}$$

We have added the above detailed parameterization (including frequencies and exponential rates) to the revised manuscript, please see the revised version, page 22, lines 535th-538th in the caption of Fig.4.

In addition, professor Holme also provided a nice consideration (or guidance) for our future work, we would like to check the prediction property of our numerical exponential model in future using a longer observed LOD data. Thank professor Holme very much once again.